# ThinkSound: Chain-of-Thought Reasoning in Multimodal Large Language Models for Audio Generation and Editing

**Huadai Liu**[1,2]**, Kaicheng Luo**[2]**, Jialei Wang**[3]**, Wen Wang**[2]
**Qian Chen**[2]**, Zhou Zhao**[3]**, Wei Xue**[1] *
[1]Hong Kong University of Science and Technology (HKUST)
[2]Tongyi Fun Team, Alibaba Group    [3]Zhejiang University

## Abstract

While end-to-end video-to-audio generation has greatly improved, producing high-fidelity audio that authentically captures the nuances of visual content remains challenging. Like professionals in the creative industries, this generation requires sophisticated reasoning about items such as visual dynamics, acoustic environments, and temporal relationships. We present **ThinkSound**, a novel framework that leverages Chain-of-Thought (CoT) reasoning to enable stepwise, interactive audio generation and editing for videos. Our approach decomposes the process into three complementary stages: foundational foley generation that creates semantically coherent soundscapes, interactive object-centric refinement through precise user interactions, and targeted editing guided by natural language instructions. At each stage, a multimodal large language model generates contextually aligned CoT reasoning that guides a unified audio foundation model. Furthermore, we introduce **AudioCoT**, a comprehensive dataset with structured reasoning annotations that establishes connections between visual content, textual descriptions, and sound synthesis. Experiments demonstrate that ThinkSound achieves state-of-the-art performance in video-to-audio generation across both audio metrics and CoT metrics, and excels in the out-of-distribution Movie Gen Audio benchmark. The project page is available at `https://ThinkSound-Project.github.io`.

## 1 Introduction

Generating realistic sound for video demands more than recognizing objects; it requires reasoning about complex visual dynamics and context – determining when an owl is chirping versus flapping its wings, identifying the subtle sway of tree branches, and synchronizing multiple sound events within a scene, as illustrated in Figure 1. Current end-to-end video-to-audio (V2A) generation systems (Luo et al., 2024; Xing et al., 2024; Zhang et al., 2024), while having largely improved, often struggle with this compositional complexity and contextual nuance. They may produce generic sounds or fail to synchronize precisely with subtle visual cues, thus limiting fidelity and user control.

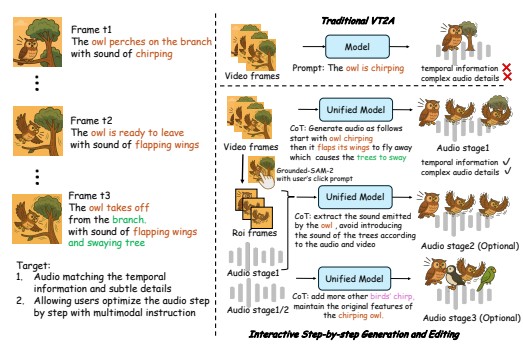

Figure 1: ThinkSound with CoT: (1) CoT-driven foley synthesis captures semantic and temporal details (2) interactive object-centric refinement for control (3) targeted editing.

---

*Corresponding author.

39th Conference on Neural Information Processing Systems (NeurIPS 2025).

Recent breakthroughs in Multimodal Large Language Models (MLLMs) (Lu et al., 2024; Liu et al., 2023b; Wang et al., 2024b) offer powerful capabilities in understanding and reasoning across multiple modalities. Concurrently, Chain-of-Thought (CoT) prompting (Wei et al., 2022; Zhang et al., 2023; Wang et al., 2025) has proven effective at eliciting structured, step-by-step reasoning from large models. These advancements create great potential to fundamentally rethink V2A by decomposing complex sound design into explicit reasoning steps and actionable synthesis instructions. The necessity for CoT in V2A becomes evident when examining the process of sound designers. They employ a multi-stage approach: analyzing visual content, then reasoning about acoustic properties, and finally synthesizing and refining sounds. End-to-end approaches compress the process into a single black-box transformation, losing the nuanced reasoning required for audio generation.

Some approaches have attempted to adopt MLLMs for multi-stage V2A. SonicVisionLM (Xie et al., 2024) converts video to textual captions using MLLMs and employs a separate model for text-to-audio (T2A) generation, which inevitably loses critical visual details and motion dynamics essential for realistic sound synthesis. More recently, DeepSound-V1 (Liang et al., 2025), while introducing MLLM-generated CoT for V2A, fragments the process into three separate tasks (audio generation, vocal removal, and silence detection) using specialized models rather than a unified framework. This fragmentation fails to fully leverage the deep understanding and reasoning capabilities that MLLMs could bring to comprehensive audio design.

To overcome these limitations and unlock the full reasoning potential of MLLMs for V2A, we present **ThinkSound** that harnesses CoT reasoning to enable stepwise, interactive generation and editing for V2A. ThinkSound decomposes audio generation into three intuitive and user-centric stages: (1) *foundational foley generation* to synthesize semantic and temporal matching soundscapes, (2) *interactive object-centric refinement* via user clicks, and (3) *targeted audio editing* guided by high-level natural language instructions. At each stage, an MLLM produces semantically and temporally aligned CoT instructions that guide a unified foundation model for audio generation, ensuring that the generated audio remains coherent, contextually grounded, and high quality throughout the workflow. Moreover, to support the training of ThinkSound and advance research in this area, we introduce **AudioCoT**, a comprehensive large-scale dataset with structured CoT reasoning annotations.

Technically, three key innovations are proposed: a) We fine-tune MLLMs on AudioCoT to generate structured, audio-specific reasoning chains that explicitly capture temporal dependencies, acoustic properties, and the decomposition of complex audio events. b) We design a unified audio foundation model based on flow matching that supports all three stages with the capabilities of synthesizing high-fidelity audio from arbitrary combinations of input modalities. This foundation model directly benefits from the detailed CoT reasoning provided by MLLMs, which effectively decomposes complex audio scenes into manageable components, enabling focused sound event generation while maintaining global coherence. c) We introduce a novel click-based interface that enables users to target specific visual objects for audio refinement, with CoT reasoning translating visual attention into contextually appropriate sound synthesis. The main contributions are summarized as follows:

- A novel three-stage interactive framework for V2A that progressively builds soundscapes through initial generation, object-centric refinement, and targeted editing, all unified by CoT reasoning from MLLMs.
- A unified multimodal foundation model capable of high-quality audio synthesis from an arbitrary combination of video, text, and audio inputs, leveraging CoT instructions to decompose complex scenes into manageable sound components.
- AudioCoT, a large-scale multimodal dataset with audio-specific CoT reasoning annotations that bridges visual content, textual descriptions, and sound synthesis.
- Experimental results demonstrate that ThinkSound achieves the state-of-the-art performance across objective metrics and subjective metrics, highlighting the effectiveness of our reasoning-guided generation.

## 2 Related Work

### 2.1 Video-to-Audio Generation

V2A (Cheng et al., 2024a; Wang et al., 2024c; Xu et al., 2024; Liu et al., 2025) focuses on synthesizing audio that aligns seamlessly with the visual content of a video clip. Earlier work (Luo et al., 2024;

Xing et al., 2024; Zhang et al., 2024; Viertola et al., 2024) try to generate audio samples based on silent video only using latent diffusion models (Rombach et al., 2022) or language models (Floridi & Chiriatti, 2020). For example, Diff-Foley (Luo et al., 2024) employs an audio-visual contrastive feature and latent diffusion to predict the spectrogram latent, while FoleyGen uses autoregressive techniques. However, recent research (Chen et al., 2024; Mo et al., 2024; You et al., 2025) pay more attention to generating audios using multimodal control, including videos, audios, and texts. For example, MMAudio (Polyak et al., 2024) uses flow matching conditioned on multi-modal inputs, including videos and texts. MultiFoley (Wu et al., 2024) further uses audio context as an additional input. Despite these advancements, existing work struggles to reason beyond simple object recognition and fails to conduct a step-by-step interactive process with users for audio generation. In this work, we propose ThinkSound, a V2A model with CoT reasoning in MLLMs, which supports step-by-step and interactive audio generation and editing.

## 2.2 Large Language Models and Reasoning

LLMs (Liu et al., 2024a; Guo et al., 2025; Hurst et al., 2024) have demonstrated remarkable reasoning capabilities through CoT prompting, enabling complex problem decomposition via intermediate reasoning steps. This paradigm, pioneered by (Wei et al., 2022), has been extended to MLLMs (Alayrac et al., 2022; Yang et al., 2023; Chu et al., 2024) that integrate visual, audio, and textual understanding through cross-modal alignment. Recent works (Rubenstein et al., 2023; Li et al., 2023; Wu et al., 2024) have explored MLLMs' potential in multimodal reasoning, particularly in visual-audio-textual grounding and cross-modal causal reasoning. Despite these, MLLMs remain under-explored for audio generation. While models like SonicVisionLM (Xie et al., 2024) and DeepSound-V1 (Liang et al., 2025) incorporate V2A, they lack mechanisms to decompose user intent into semantic and temporal reasoning steps. Our ThinkSound proposes a novel framework for audio generation and editing that decomposes the complex task into foundation Foley generation, interactive object-centric refinement, and targeted audio editing (Wang et al., 2023; Liu et al., 2024d).

## 2.3 Flow Matching

Flow Matching has emerged as a powerful alternative to diffusion models for high-quality audio synthesis (Evans et al., 2024; Liu et al., 2024b, 2023a), directly learning continuous normalizing flows between distributions by training a time-dependent vector field. Recent works Lipman et al. (2022); Liu et al. (2022) establish its theoretical foundations, while Liu et al. (2024c) demonstrated its advantages for audio generation with improved quality and faster sampling. Polyak et al. (2024) further advances this approach by applying conditional flow matching to multimodal audio generation, showing its effectiveness in maintaining temporal coherence across complex audio signals. Our work extends these advances through CoT-guided generation and a novel multi-guidance strategy that integrates control signals from multiple modalities. Our modality-agnostic training approach with classifier-free guidance dropout allows the model to generate high-quality audio from flexible input combinations, addressing both control precision and data scarcity challenges that have limited previous approaches.

# 3 AudioCoT Dataset for CoT-Guided Generation and Editing

## 3.1 Multimodal Data Sources

The AudioCoT dataset comprises both video-audio and audio-text pairs. For video-audio data, we utilize VGGSound (Chen et al., 2020) and a curated, non-speech subset of AudioSet (Gemmeke et al., 2017) to ensure broad coverage of real-world audiovisual events. For audio-text data, we aggregate pairs from AudioSet-SL (Hershey et al., 2021), Freesound (Fonseca et al., 2017), AudioCaps (Kim et al., 2019), and BBC Sound Effects [2], resulting in a diverse and representative corpus for training multimodal models. Further details on data processing and statistics are provided in the Appendix A.1.

---

[2]https://sound-effects.bbcrewind.co.uk/

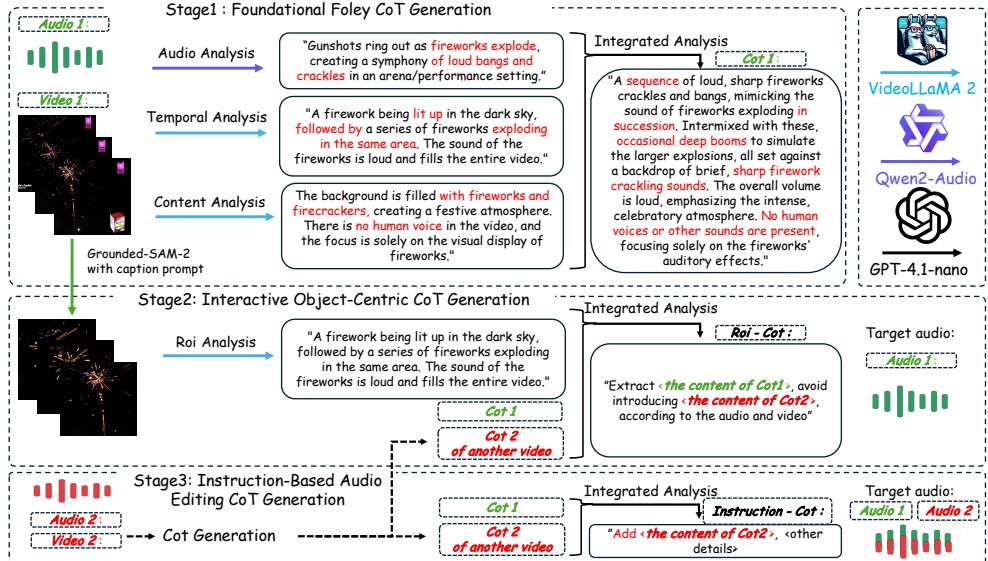

Figure 2: Overview of our AudioCoT dataset construction pipeline.

## 3.2 Automated CoT Generation Pipeline

**Stage 1: Foundational Foley CoT Generation.** To construct the Foley generation CoT data, we employ a multi-stage processing pipeline. For video-audio pairs, we utilize VideoLLaMA2 (Cheng et al., 2024b) to extract both temporal and semantic information from the videos through different prompting strategies, while for audio-text pairs (which lack video data), we employ a more streamlined approach without VideoLLaMA 2. In both cases, we generate audio descriptions using Qwen2-Audio (Chu et al., 2024), combining them with existing video captions or text annotations. The collected information is then integrated using GPT-4.1-nano[3] to synthesize comprehensive CoT reasoning chains that capture the complex relationships between content conditions and the corresponding audio elements, ensuring both data types contribute to a comprehensive understanding of audio generation reasoning.

**Stage 2: Interactive Object-Centric CoT Generation.** To facilitate object-focused audio generation, we develop a Region of Interest (ROI) extraction framework leveraging Grounded SAM2 (Ren et al., 2024; Ravi et al., 2024). Using audio captions as prompts, we identify and generate bounding boxes for potential sound-emitting objects. These coordinates are tracked temporally across video frames, while VideoLLaMA2 provides detailed semantic descriptions for each ROI segment. For complex audio manipulations (extraction/removal), we construct a hierarchical reasoning structure where the CoT of the target video is merged with another video's CoT to establish a global context. This composite representation is then integrated with ROI-specific generation information and processed by GPT-4.1-nano to formulate coherent manipulation rationales (detailed in Appendix A.2).

**Stage 3: Instruction-Based Audio Editing CoT Generation.** For instruction-guided audio editing, we analyze and integrate CoT information from Stage 1 based on four primary operations: extension, inpainting, addition, and removal. These operations address scenarios from extending sequences to eliminating unwanted segments. GPT-4.1-nano processes this integrated information to generate instruction-specific CoT reasoning chains while we perform corresponding audio operations, creating (Instruction-CoT, input audio, output audio) triplets for model training and evaluation.

## 4 ThinkSound

### 4.1 Overview

As depicted in Figure 3, ThinkSound introduces a novel step-by-step, interactive framework for audio generation and editing guided by CoT reasoning. Our approach decomposes the complex V2A task into three intuitive stages: (1) foundation Foley generation that creates a semantic and temporal

---

[3]https://openai.com/index/gpt-4-1/

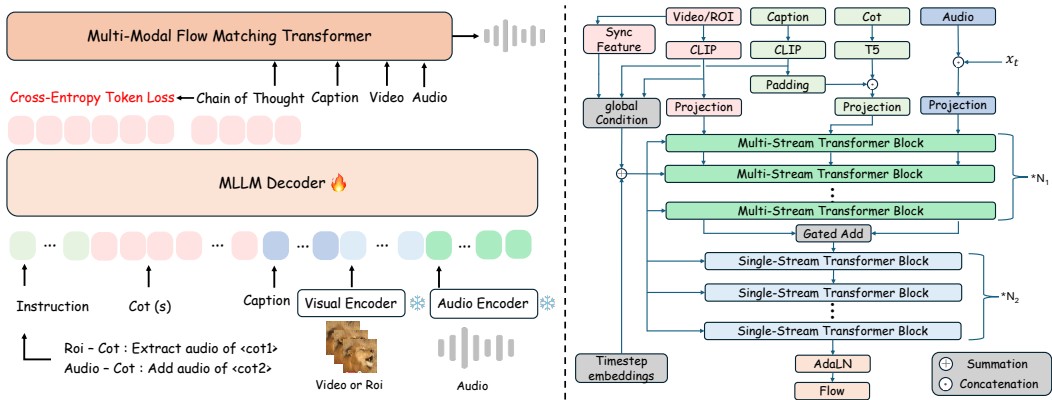

Figure 3: Overview of the ThinkSound architecture. **Left:** our Multimodal LLM framework, where a fine-tuned VideoLLaMA 2 model generates CoT reasoning for audio generation and editing. **Right:** our enhanced Multimodal Transformer architecture, which features an MM-DiT backbone with dedicated pathways for processing multimodal inputs and CoT-driven conditioning to enable high-fidelity, contextually grounded audio generation.

matching soundscape, (2) interactive region-based refinement through user clicks, and (3) targeted audio editing based on high-level instructions. At each stage, an MLLM generates CoT reasoning that guides a unified audio foundation model to produce and refine the soundtrack.

## 4.2 CoT Reasoning with Multimodal LLM

To enable stepwise, context-aware audio generation, we leverage VideoLLaMA2 (Cheng et al., 2024b) as the core multimodal reasoning engine. VideoLLaMA2 is selected for its state-of-the-art capability in fusing video, text, and audio modalities, and its advanced spatial-temporal modelling is essential for capturing the nuanced interplay between visual events and their corresponding acoustic manifestations.

We further adapt VideoLLaMA2 to the audio reasoning domain by fine-tuning it on our AudioCoT dataset, which contains rich, annotated reasoning chains tailored for audio-visual tasks. This fine-tuning process is designed to equip the model with three core competencies: (1) **audio-centric understanding**–the ability to infer acoustic properties, model sound propagation, and reason about audio-visual correspondences, including temporal and causal relationships among audio events (e.g., "footsteps occur before the door opens, then conversation ensues"); (2) **structured CoT decomposition**–the capacity to break down complex audio generation or editing tasks into explicit, actionable steps; and (3) **multimodal instruction following**–robustly interpreting and executing diverse generation or editing instructions across modalities. As illustrated in Figure 3, the fine-tuning objective is the standard cross-entropy loss for next-token prediction. Through this targeted adaptation, VideoLLaMA2 is transformed into a specialized audio reasoning module, capable of producing contextually precise CoT instructions that drive each stage of the ThinkSound pipeline.

## 4.3 CoT-Guided Unified Audio Foundation Model

The core of ThinkSound is our unified audio foundation model that seamlessly translates CoT reasoning into high-quality audio, whose detail is shown in the right part of Figure 3. We encode audio into latent representations using a pre-trained VAE (Pinheiro Cinelli et al., 2021) and train our model with conditional flow matching (Lipman et al., 2022), where the velocity field prediction is conditioned on multimodal context, including visual content, CoT reasoning, text captions, and audio context. To support any combination of input modalities, we adopt the integration of classifier-free guidance dropout during training. By randomly dropping different modality combinations with probability $p_{\mathrm{drop}}$, we enable the model to handle arbitrary input configurations at inference time—essential for our interactive framework. We also incorporate strategic audio context masking to support advanced editing operations such as inpainting and extension.

For text processing, we employ a dual-pathway encoding strategy: MetaCLIP (Xu et al., 2023) encodes visual captions to provide scene-level context, while T5-v1-xl (Raffel et al., 2020) processes structured CoT reasoning to capture detailed temporal and causal relationships. These complementary

representations are effectively combined, with MetaCLIP's features serving as global conditioning signals while T5's outputs enable fine-grained reasoning-driven control.

Our enhanced MM-DiT architecture builds on recent advances in multimodal generative modeling (Esser et al., 2024; Labs, 2024; Cheng et al., 2024a) with three key components: (1) we implement a hybrid transformer backbone that alternates between modality-specific and shared processing. Multi-stream transformer blocks maintain separate parameters for each modality while sharing attention mechanisms, allowing efficient processing of diverse inputs without sacrificing cross-modal learning. (2) We design an adaptive fusion module that upsamples video features and fuses them with audio latents via a gated mechanism (Cho et al., 2014). This not only highlights salient visual cues and suppresses irrelevant information, but also ensures that video information is directly involved in subsequent single-stream transformer blocks. By integrating video into the audio latent space, the model can better capture subtle visual details and their nuanced effects on the soundscape, enabling richer cross-modal reasoning than using audio latents alone. (3) We implement global conditioning by mean-pooling CLIP features from both the caption and video, and, following MMAudio (Cheng et al., 2024a), incorporating sync features to improve audio-visual temporal alignment. The resulting global condition is added to the timestep embedding and injected via adaptive layer normalization layers (AdaLN) (Peebles & Xie, 2023) into both multi-stream and single-stream blocks.

### 4.4 Step-by-Step CoT-Guided Audio Generation and Editing

By enabling flexible combinations of input modalities along with CoT, ThinkSound supports decomposing audio generation into three intuitive stages shown in Figure 1. This three-stage pipeline enables progressively refined, highly customized audio generation through an intuitive interactive workflow, with CoT reasoning bridging user intent and audio synthesis at each step.

**Stage 1: CoT-Guided Foley Generation.**   In the first stage, our framework analyzes the entire video to identify acoustic elements and their relationships. The fine-tuned MLLM generates detailed CoT reasoning that explicitly identifies primary sound events, ambient elements, acoustic properties, and their temporal dependencies - determining when objects make sounds and how these sounds interact. This structured reasoning guides the audio foundation model to synthesize high-fidelity audio that precisely matches both the semantic content and temporal dynamics of the visual scene. By decomposing complex audio scenes into explicit sound components through CoT reasoning, the model generates a diverse and coherent soundscape that captures subtle visual cues and motion dynamics essential for realistic audio synthesis.

**Stage 2: Interactive Object-Focused Audio Generation.**   Stage 2 introduces an interactive framework that enables users to refine the initial soundscape by focusing on specific visual elements. Through a simple click-based interface, users can select objects of interest for audio enhancement. Unlike the holistic approach in Stage 1, this object-centric refinement leverages the segmented ROI to guide targeted audio synthesis. The fine-tuned MLLM analyzes the selected ROI and generates specialized CoT reasoning focused on the object's acoustic properties within the global context. This structured reasoning conditions the audio foundation model to synthesize object-specific sounds, seamlessly integrating them with the initial soundtrack produced in Stage 1. Notably, in this stage, the foundation model incorporates the existing audio context as an additional conditioning signal.

**Stage 3: Instruction-based Audio Editing.**   In the final stage, users can provide high-level editing instructions to refine audio quality or modify specific elements. The MLLM translates these natural language instructions into precise audio processing operations through CoT reasoning, considering both visual content and current audio state. The foundation model, conditioned on both this reasoning and the existing audio context, applies targeted modifications while maintaining overall coherence. This natural language-driven approach enables non-technical users to perform sophisticated audio manipulation, including adding sounds, removing sounds, audio inpainting, and audio extension.

## 5 Experiments

### 5.1 Experiment Setup

**Evaluation Metrics** We conduct comprehensive evaluations using both objective and subjective metrics to assess audio quality, text-audio alignment, and video-audio synchronization. For objective

Table 1: Comparison of our ThinkSound foundation model with existing video-to-audio baselines on the VGGSound test set. ↓ indicates lower is better, ↑ indicates higher is better. For MOS, we show the mean and variance of the MOS scores. $\dagger$ indicates that the method does **not use text** for inference.

| Method | Objective Metrics | | | | | | Subjective Metrics | | Efficiency | |
| --- | --- | --- | --- | --- | --- | --- | --- | --- | --- | --- |
| | FD↓ | KL$_{PaSST}$ ↓ | KL$_{PaNNs}$ ↓ | DeSync↓ | CLAP$_{cap}$ ↑ | CLAP$_{CoT}$ ↑ | MOS-Q↑ | MOS-A↑ | Params | Time(s)↓ |
| GT | - | - | - | 0.55 | 0.28 | 0.45 | 4.37±0.21 | 4.56±0.19 | - | - |
| See&Hear | 118.95 | 2.26 | 2.30 | 1.20 | 0.32 | 0.35 | 2.75±1.08 | 2.87±0.99 | 415M | 19.42 |
| V-AURA$\dagger$ | 46.99 | 2.23 | 1.83 | 0.65 | 0.23 | 0.37 | 3.42±1.03 | 3.20±1.17 | 695M | 14.00 |
| FoleyCrafter | 39.15 | 2.06 | 1.89 | 1.21 | **0.41** | 0.34 | 3.08±1.21 | 2.63±0.88 | 1.20B | 3.84 |
| Frieren$\dagger$ | 74.96 | 2.55 | 2.64 | 1.00 | 0.37 | 0.34 | 3.27±1.11 | 2.95±1.09 | 159M | - |
| V2A-Mapper$\dagger$ | 48.10 | 2.50 | 2.34 | 1.23 | 0.38 | 0.32 | 3.31±1.02 | 3.16±1.04 | 229M | - |
| MMAudio | 43.26 | 1.65 | 1.40 | **0.44** | 0.31 | 0.40 | 3.84±0.89 | 3.97±0.82 | 1.03B | 3.01 |
| **ThinkSound** | **34.56** | **1.52** | **1.32** | 0.46 | 0.33 | **0.46** | **4.02±0.73** | **4.18±0.79** | 1.30B | 1.07 |
| w/o CoT Reasoning | 39.84 | 1.59 | 1.40 | 0.48 | 0.29 | 0.41 | 3.91±0.83 | 4.04±0.75 | 1.30B | 0.98 |

evaluation, we compute the **Fréchet Distance (FD)** (Kilgour et al., 2018) in the OpenL3 feature space (Cramer et al., 2019; Evans et al., 2024) to measure distribution-level similarity, which we extend to evaluate stereo audio. We also use the **Kullback-Leibler (KL) Divergence** (Copet et al., 2024) based on predictions from PaSST model (KL$_{PaSST}$) and PaNNs (KL$_{PaNNs}$) to evaluate label-level consistency. Temporal alignment between audio and video is assessed using **DeSync** predicted by the Synchformer model (Iashin et al., 2024), while semantic alignment with text is measured using the **CLAP score** (Wu* et al., 2023; Chen et al., 2022), including both CLAP$_{cap}$ (caption) and CLAP$_{CoT}$ (CoT). For subjective evaluation, we collect human ratings using the **Mean Opinion Score (MOS)** to evaluate perceived audio quality (**MOS-Q**) and alignment with video and CoT (**MOS-A**). Further details are provided in the Appendix C.

**Implementation Details** For VAE training, we initialize our VAE using the VAE model weights trained on stereo data at 44.1kHz sample rate provided by Stability AI [4]. We employ mixed precision training with a batch size of 144 across 24 A800 GPUs for 500,000 steps. Subsequently, following Evans et al. (2024), we freeze the VAE encoder and train the VAE decoder with a latent mask ratio of 0.1 for an additional 500,000 steps. We use AdamW (Loshchilov & Hutter, 2019) as the optimizer, setting the generator learning rate to 3e-5 and the discriminator learning rate to 6e-5. In the foundation model training phase, we utilize an exponential moving average and automatic mixed precision for 100,000 steps on 8 A100 GPUs, with an effective batch size of 256. We adopt a cfg dropout of 0.2 for each modality with a learning rate of 1e-4. During the task-specific fine-tuning stage, we similarly apply exponential moving average and automatic mixed precision for 50,000 steps on 8 A100 GPUs, maintaining an effective batch size of 256. AdamW remains our optimizer of choice, with a learning rate set at 1e-4. We attach the benchmark details of VGGSound, Movie Gen Audio Bench, and AudioCoT test set for stages 2 and 3 into Appendix A.3.

## 5.2 Main Results

**Video-to-Audio Generation Results** We compare our foundation model against existing video-to-audio baselines including Seeing and Hearing (Xing et al., 2024), V-AURA (Viertola et al., 2024), FoleyCrafter (Zhang et al., 2024), Frieren (Wang et al., 2024d), V2A-Mapper (Wang et al., 2024a), and MMAudio (Cheng et al., 2024a). From Table 1, we observe three key findings: (1) GT audio exhibits low CLAP$_{cap}$ scores (0.28), revealing that original VGGSound captions inadequately capture the semantic content and temporal relationships needed for high-fidelity audio generation. (2) ThinkSound outperforms all baselines across most objective metrics and all subjective metrics. Compared to the strongest baseline (MMAudio), our model achieves substantial improvements in audio quality (KL$_{PaSST}$: 1.52 vs. 1.65) and semantic alignment (CLAP$_{CoT}$: 0.46 vs. 0.40), while maintaining comparable temporal synchronization. Subjective evaluations further confirm these improvements (MOS-Q: 4.02 vs. 3.84, MOS-A: 4.18 vs. 3.97). (3) Removing CoT reasoning notably degrades both audio quality and alignment metrics, especially for CLAP$_{CoT}$, decreasing from 0.46 to 0.41, confirming that CoT provides crucial information about sound events, their temporal relationships, and acoustic characteristics.

Table 2: Out-of-distribution evaluation on MovieGen Audio Bench. This benchmark does not provide the GT audios, so we cannot compare FD and KL.

| Method | CLAP$_{CoT}$ ↑ | DeSync↓ | MOS-Q↑ | MOS-A↑ |
| --- | --- | --- | --- | --- |
| MMAudio | 0.45 | 0.77 | 3.95±0.87 | 3.62±1.03 |
| MovieGen | 0.47 | 1.00 | 3.98±0.77 | 3.70±0.96 |
| ThinkSound | **0.51** | **0.76** | **4.11±0.74** | **3.87±0.82** |

[4]https://github.com/Stability-AI/stable-audio-tools

Furthermore, we conduct an out-of-distribution evaluation on the MovieGen Audio Bench (Polyak et al., 2024) to assess the generalization capability of our model. The results in Table 2 show that our ThinkSound model still achieves the best $CLAP_{CoT}$ score of 0.51 while DeSync is on par with the best baseline. For subjective evaluation, ThinkSound achieves the best performance in both alignment and fidelity metrics. This demonstrates that ThinkSound exhibits strong generalization capability across different scenarios.

**Object-Focused Audio Generation and CoT-Guided Audio Editing** For object-focused generation, we compare two approaches: (1) MMAudio, which does not have the ROI design, and (2) ThinkSound w/o CoT reasoning. Results in Table 3 show that our interactive approach with CoT reasoning achieves significantly better results, demonstrating superior object-specific sound quality and integration with the foundation audio.

Table 3: Object-focused generation performance.

| Method | FD↓ | $KL_{PaSST}$ ↓ | CLAP↑ | MOS-Q↑ | MOS-A↑ |
|---|---|---|---|---|---|
| MMAudio | 44.46 | 1.38 | 0.41 | 3.61±0.63 | 3.64±0.69 |
| ThinkSound | **43.27** | **1.32** | **0.48** | **3.89±0.52** | **3.91±0.53** |
| w/o CoT | 45.28 | 1.34 | 0.43 | 3.77±0.64 | 3.81±0.59 |

For text-guided audio editing, we compare ThinkSound with AudioLDM-2 (Liu et al., 2024e) and Edit Friendly DDPM (Huberman-Spiegelglas et al., 2024), adapting these models to our experimental setup. As shown in Table 4, ThinkSound outperforms the baselines across all objective and subjective metrics. Specifically, ThinkSound achieves the lowest FD (34.78) and $KL_{PaSST}$ (1.45), and the highest CLAP score (0.51), indicating better audio fidelity and se-

Table 4: Audio editing results on AudioCoT test set (MOS-A: alignment between audio and text; DDPM: DDPM-Friendly).

| Method | FD↓ | $KL_{PaSST}$ ↓ | CLAP↑ | MOS-Q↑ | MOS-A↑ |
|---|---|---|---|---|---|
| AudioLDM-2 | 61.28 | 1.94 | 0.35 | 3.28±0.59 | 3.48±0.82 |
| DDPM | 55.56 | 1.75 | 0.39 | 3.34±0.28 | 3.67±0.56 |
| ThinkSound | **34.78** | **1.45** | **0.51** | **3.92±0.82** | **3.85±0.82** |
| w/o CoT | 45.78 | 1.58 | 0.44 | 3.53±0.45 | 3.52±0.62 |

mantic relevance. In human evaluation, ThinkSound also gets the best MOS-Q and MOS-A. Removing CoT reasoning clearly drops all metrics, showing the importance of CoT-guided reasoning for text-based audio editing.

## 5.3 Ablation Studies

To better understand the contribution of each component in ThinkSound and to validate the effectiveness of our design choices, we conduct comprehensive ablation studies on the VG-GSound test set. Mainly focus on: 1) text encoding strategies and 2) multi-modal integration mechanisms. For more ablation and exploratory results, we attach them to Appendix D.

**Text Encoding Strategies.** We evaluate different text encoding strategies with or without CoT reasoning, and the results are compiled in Table 5. The results show that (1) CoT reasoning substantially improves audio fidelity, with the FD score improving from 39.84 to 37.65 when comparing CLIP-only to T5-based CoT approaches. (2) The integration of contrastive features from CLIP with contextual features from T5 further enhances performance, reducing KL

Table 5: Comparison of text encoder fusion strategies. The input of the CLIP text encoder is the caption. The CLAP score means the value of $CLAP_{CoT}$.

| Method | FD↓ | $KL_{PaSST}$ ↓ | $KL_{PaNNs}$ ↓ | DeSync↓ | CLAP ↑ |
|---|---|---|---|---|---|
| CLIP | 39.84 | 1.59 | 1.40 | 0.48 | 0.41 |
| T5 (CoT) | 37.65 | 1.54 | 1.35 | 0.46 | 0.44 |
| CLIP+T5 | **34.56** | **1.52** | **1.32** | **0.46** | **0.46** |

divergence metrics from 1.54 to 1.52 ($KL_{PaSST}$) and from 1.35 to 1.32 ($KL_{PaNNs}$).

**Multi-Modal Integration Mechanisms.** We investigate different multi-modal integration mechanisms for video and audio before feeding them into the single-stream transformer, and the results are displayed in Table 6. We find that (1) the element-wise addition of video and audio features performs better than audio-only input with weak-supervision global conditioning, especially for synchronization metric DeSync,

Table 6: Comparison of different multi-modal integration mechanisms between video and audio features.

| Integration | FD↓ | $KL_{PaSST}$ ↓ | $KL_{PaNNs}$ ↓ | DeSync↓ | CLAP ↑ |
|---|---|---|---|---|---|
| audio only | 37.13 | 1.58 | 1.37 | 0.50 | 0.43 |
| linear video | 38.96 | 1.58 | 1.38 | 0.46 | 0.45 |
| gated video | **34.56** | **1.52** | **1.32** | **0.46** | **0.46** |

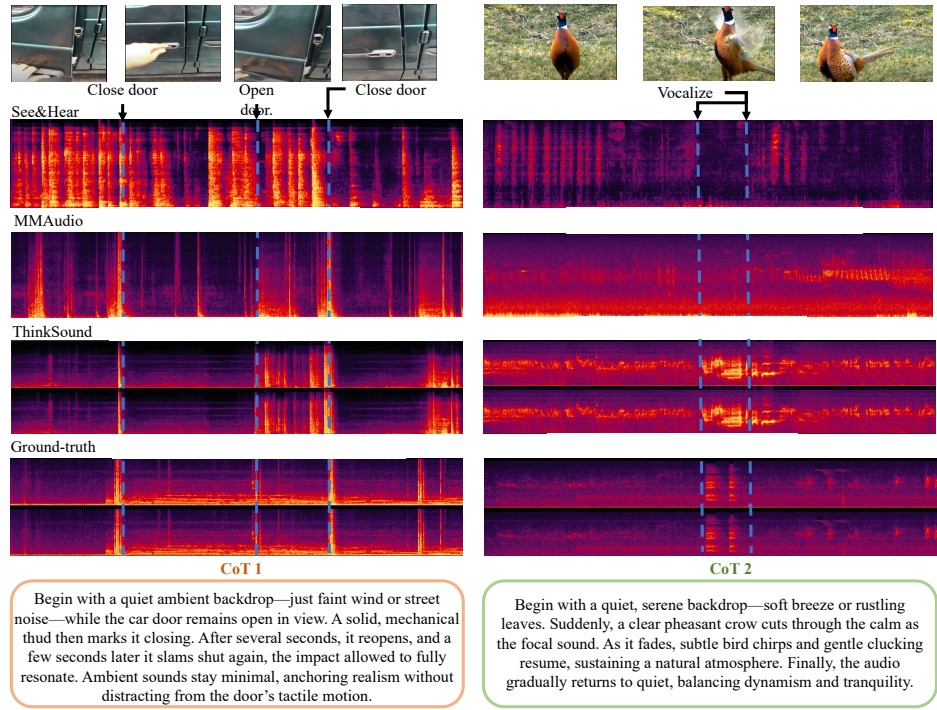

Figure 4: Qualitative Comparison: **Left:** Spectrograms for a car door movement sequence (closed → opened → closed), showing ThinkSound's precise alignment of each door sound versus the baseline's premature opening effect. **Right:** Spectrograms for a grassy-field pheasant scene (ambient bird calls → wing-flap chirp → ambient calls), illustrating ThinkSound's accurate detection and timing of the transient chirp compared to the baseline's omission or delay.

decreasing from 0.50 to 0.46. (2) The gated fusion mechanism outperforms simple element-wise addition and audio-only input across all metrics.

## 5.4 Case Study

In our qualitative analysis, we compare spectrograms of audio generated by ThinkSound and those produced by baselines, as illustrated in Figure 4. We make the following observations: (1) As demonstrated in case 1, a car door sequence—closed–opened–closed—is accurately reflected by ThinkSound, whereas the baseline models erroneously introduce an extra opening sound at the start. This highlights ThinkSound's ability, via CoT prompting, to track temporal and causal event order in structured scenes. (2) In case 2, a pheasant moving in a grassy field—accompanied by ambient bird calls, suddenly flapping its wings and emitting a sharp chirp before returning to background noise—is faithfully reproduced by ThinkSound. The baselines, however, often miss or delay this brief chirp. These cases underscore ThinkSound's enhanced temporal reasoning and sensitivity to subtle visual cues, resulting in more precise and context-aware audio synthesis.

## 6 Conclusion

This paper presented ThinkSound, a novel CoT reasoning framework for audio generation and editing that decomposed complex audio generation into three interpretable and user-centric stages. Our evaluations showed that ThinkSound outperformed state-of-the-art methods, producing contextually appropriate and temporally precise soundscapes. The framework's interactive nature enabled users to refine generated audio through intuitive pointing and natural language instructions, addressing the gap between creative intent and automated generation. While challenges remained in capturing nuanced acoustic properties, our AudioCoT dataset and training strategy established a foundation for future research in intelligent audio generation with applications across film, gaming, and social media. In future work, we will explore incorporating physical acoustic modeling and developing more sophisticated reasoning capabilities for complex multi-object sound interactions.

## Acknowledgements

The research was supported by the NSFC (No. 62206234) of Mainland China.

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

# A AudioCoT Dataset Details

## A.1 Data Collection and Preprocessing

To ensure high data quality and consistency, we employ a comprehensive preprocessing pipeline. We begin by removing silent audio-video clips to retain only those with meaningful content. For the AudioSet subset (Gemmeke et al., 2017), we further exclude segments containing human voices based on tag information, as our focus is on non-speech audio. All audio-video clips are then segmented into fixed-length intervals of 9.1 seconds, with any shorter clips discarded to maintain uniformity. To achieve a balanced dataset, we maintain an approximately 1:1 ratio between music and sound effect samples, ensuring equal representation of both categories.

## A.2 Quality Control for Automated Data Pipeline

To enhance the effectiveness of CoT reasoning in preserving audio characteristics while integrating visual context, we implemented a comprehensive multi-stage quality control pipeline:

**Stage 1: Audio-Text Alignment Filtering**   We employ a systematic approach to ensure high-quality audio-CoT pairs. First, we calculate the CLAP score between each audio sample and its corresponding CoT description to quantify semantic alignment. For pairs exhibiting low CLAP scores (below 0.2), we regenerate the CoT using an enhanced prompt specifically designed to emphasize audio characteristics and features. After regeneration, we recalculate the CLAP score to assess improvement. Audio samples that continue to demonstrate poor alignment (persistent low CLAP scores) are excluded from the dataset to maintain quality standards.

**Stage 2: Object Tracking Consistency**   To ensure reliable audio-visual correspondence, we retain only those video sequences containing at least one Region of Interest (ROI) that remains consistently visible throughout the entire duration. Videos with objects that disappear from view or exhibit inconsistent tracking are filtered out. This criterion ensures that our dataset maintains high-quality visual references for audio generation tasks, providing consistent visual anchors for the audio generation process.

**Stage 3: Semantic Pairing of Audio Components**   For tasks requiring paired audio components, we utilize GPT-4.1-nano to analyze tag categories from VGGSound (Chen et al., 2020) based on two critical criteria. First, we ensure semantic distinctiveness, where tags must be sufficiently distinct to avoid confusion during audio extraction and removal tasks. Second, we verify contextual plausibility, ensuring that the co-occurrence of paired sounds is contextually reasonable within the same acoustic scene. This balanced approach ensures that our audio pairs are both semantically meaningful and practically useful for audio generation tasks.

**Human Verification Protocol**   To validate our automated filtering processes, we implement a rigorous manual review at each pipeline stage. No less than 5% of the total data volume undergoes human inspection to ensure quality. This verification step helps validate the effectiveness of our automated filtering criteria and ensures the overall reliability and quality of our dataset. The human reviewers assess both the technical aspects of alignment and the perceptual quality of the audio-visual correspondence. When samples fail human verification, they are immediately removed from the dataset. Additionally, if the human rejection rate for any filtering criterion exceeds 5%, we recalibrate the corresponding automated filtering parameters and reprocess the entire batch to maintain dataset integrity. This feedback loop between automated filtering and human verification ensures continuous improvement of our quality control pipeline.

## A.3 Benchmark Construction

We evaluate the performance of ThinkSound on three different tasks: video-to-audio generation, object-focused audio generation, and audio editing. For the video-to-audio generation task, we use the VGGSound test set as the in-distribution evaluation set while the MovieGen Audio Bench is the out-of-distribution evaluation set. For the VGGSound test set, we use the same quality filtering protocol as our training data preparation. Given that our primary focus is on video-to-sound/music generation, we construct three different difficulty levels based on the complexity of the audio-visual

Table 7: Overview of datasets used in our work.

| Dataset | Modality | Text Format | Hours |
|---|---|---|---|
| VGGSound (Chen et al., 2020) | Audio-Video | Caption | **453.6** |
| AudioSet (Gemmeke et al., 2017) | Audio-Video | Caption | 287.5 |
| AudioSet-SL (Hershey et al., 2021) | Audio-Text | Caption | 262.6 |
| Freesound (Fonseca et al., 2017) | Audio-Text | Caption | **1286.6** |
| AudioCaps (Kim et al., 2019) | Audio-Text | Caption | 112.6 |
| BBC Sound Effects [5] | Audio-Text | Tags | 128.9 |
| **Total Hours** | – | – | **2531.8** |

relationships. Specifically, we distinguish the difficulty levels based on a multi-dimensional scoring system that evaluates:

- **Semantic Consistency**: The alignment between the audio and the visual content is evaluated by the ImageBind score (0.3+ for easy, 0.25-0.3 for medium, 0.2-0.25 for hard), and the CLAP score between audio and CoT (0.4+ for easy, 0.3-0.4 for medium, 0.2-0.3 for hard).
- **Temporal Synchronization**: The degree of synchronization between visual events and corresponding sounds evaluated by DeSync score (0-0.3 for easy, 0.3-0.6 for medium, 0.6+ for hard).
- **Acoustic Scene Complexity**: The audio events' numbers (one dominant sound for easy, 2-3 distinct sounds for medium, multiple overlapping sounds for hard)

According to the above evaluation criteria, we input the scores and criteria into GPT-4.1-nano to generate the difficulty level for each sample. The final difficulty assignment follows a tertile distribution: the lowest-scoring third is classified as "easy," the middle third as "medium," and the highest-scoring third as "hard." This stratified approach ensures balanced representation across difficulty levels while maintaining meaningful distinctions in task complexity. For each difficulty level, we construct a benchmark subset containing around 2000 samples.

For stages 2 and 3, we maintain methodological consistency with our training protocols while adapting the evaluation criteria to each task's specific requirements. For stage 2, we select samples with clearly identifiable visual objects that produce distinct sounds, while for stage 3, we focus on samples with different audio categories suitable for manipulation tasks. Each evaluation subset contains approximately 2,000 samples.

# B   Model Configurations and Architecture

## B.1   Model Configurations

ThinkSound consists of two primary components: a hierarchical variational autoencoder (VAE) for audio compression and reconstruction, and a flow-matching multimodal transformer.

**Variational Autoencoder**   The encoder consists of five convolutional blocks with channel multipliers [1, 2, 4, 8, 16] and strides [2, 4, 4, 8, 8], projecting the stereo waveform into a 128-dimensional latent space. The decoder mirrors this architecture with transposed convolutions to reconstruct 64-dimensional latent representations back into the time-domain waveform.

**Multimodal Diffusion Transformer**   ThinkSound employs an enhanced Multi-modal Diffusion Transformer (MM-DiT) with a hidden size dimension of 1024. It comprises 14 multi-stream transformer layers and 7 single-stream transformer layers, with 16 attention heads. We further attach our different model scale parameters for reference. We use the large model by default.

## B.2   Model Architecture

The architecture of multistream transformers is depicted in Figure 5.

Table 8: Diffusion Transformer Configurations at Different Model Sizes

| Model Scale | Hidden Size | Depth | Attention Heads | Multistream Layers | Singlestream Layers | Total Parameters |
|---|---|---|---|---|---|---|
| Large | 1024 | 21 | 16 | 14 | 7 | 1.3B |
| Medium | 768 | 21 | 12 | 14 | 7 | 724M |
| Small | 512 | 18 | 8 | 12 | 6 | 533M |

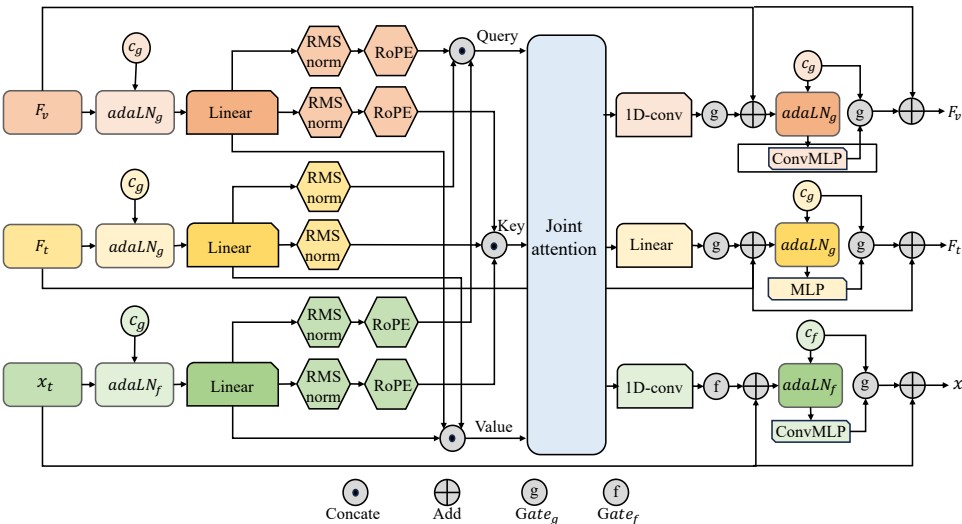

Figure 5: Multi-stream blocks: $F_v$ is the video features, $F_t$ is the text features, $x_t$ is the audio latents, and $c_g$ denotes the global condition.

## C Evaluation

### C.1 Objective Metrics

To comprehensively evaluate the generated audio, we adopt a set of objective metrics targeting different aspects: perceptual quality, semantic consistency, temporal alignment, and cross-modal correspondence.

**Feature Distribution Alignment:** We project both generated and reference audio into the OpenL3 embedding space Cramer et al. (2019); Evans et al. (2024) and compute the **Fréchet Distance (FD)** Kilgour et al. (2018); Copet et al. (2024) to assess the similarity between their distributional statistics. We chose OpenL3 because it accepts signals of up to 48kHz while VGGish operates at 16kHz, which is more suitable for our 44kHz audio. Following the previous work Evans et al. (2024), we extend the FD to evaluate stereo signals by projecting left and right channels separately and then averaging the results. Moreover, to evaluate whether the generated audio matches the reference in terms of its distribution, we compute the **Kullback-Leibler (KL) Divergence** Copet et al. (2024) between class probability distributions predicted by the PaSST model Koutini et al. (2021) and PaNNs model (Kong et al., 2020) as classifiers.

**Temporal Alignment:** To evaluate the synchronization between generated audio and its corresponding video, we adopt the **DeSync** score predicted by the Synchformer model Iashin et al. (2024), following the practice of Cheng et al. (2024a). For each sample, we truncate the video to match the duration of the generated audio and compute the DeSync score using Synchformer, which operates with a 4.8-second context window. Specifically, we extract both the first and last 4.8-second segments from each video-audio pair, calculate the DeSync score for each segment, and report the average as the final temporal alignment metric. **Text-Audio Correspondence:** To assess the semantic alignment between generated audio and textual prompts, we utilize the **CLAP score** Wu* et al. (2023); Chen et al. (2022), which measures similarity in a shared audio-text embedding space. Specifically, we report $CLAP_{cap}$ for evaluating alignment with the original video captions and $CLAP_{CoT}$ for alignment

with our constructed CoT descriptions. As discussed in Section 5.2, the original VGGSound captions are often low quality and yield lower CLAP scores, whereas our CoT annotations provide richer semantic detail and achieve higher alignment. **Consequently, we primarily use the CoT-audio alignment metric in our evaluations, except in Table 1 where caption-based alignment is also reported.**

## C.2 Subjective Metrics

Our subjective evaluation framework employs the Mean Opinion Score (MOS) methodology along two critical dimensions to comprehensively assess the generated audio:

**Audio Quality Assessment (MOS-Q)**    We evaluate the intrinsic perceptual quality of generated audio through a rigorous assessment protocol where participants are instructed to focus on four specific aspects:

- *Clarity*: The absence of unwanted artifacts, distortions, or noise

- *Naturalness*: How realistic and non-synthetic the audio sounds

- *Fidelity*: The richness and accuracy of acoustic characteristics

- *Overall impression*: The holistic listening experience

Each listener rates these qualities using a standard 5-point Likert scale (1: Poor, 2: Fair, 3: Good, 4: Very Good, 5: Excellent). The final MOS-Q score for each audio sample represents the average rating across all evaluators, providing a robust measure of perceived audio quality.

**Semantic and Temporal Alignment Assessment (MOS-A)**    To evaluate the cross-modal coherence between generated audio and visual content, we assess both semantic relevance and temporal synchronization (we also provide CoT text as the auxiliary information for semantic alignment):

- *Semantic alignment*: How well the audio content matches the objects, actions, and environment depicted in the video

- *Temporal alignment*: How accurately sound events correspond to visual events in time

Participants judge the alignment according to three categories on the same 5-point scale:

- *Fully aligned* (4-5 points): Complete semantic correspondence with precise temporal synchronization

- *Mostly aligned* (2.5-3.9 points): Good semantic match with occasional minor temporal misalignments

- *Partially aligned* (1-2.4 points): Noticeable discrepancies in either semantic content or temporal synchronization

**Evaluation Protocol**    To ensure evaluation reliability and consistency, we implemented the following protocol:

- All assessments were conducted in controlled in-person sessions with standardized audio equipment

- 15 raters with normal hearing ability were recruited and briefed on the evaluation criteria

- Each rater evaluated a random subset of 50 video-audio pairs from our test collection

- Samples were presented in randomized order to prevent ordering bias

- Reference examples of each quality level were provided before the evaluation sessions

- Raters were given sufficient time to carefully evaluate each sample

# D Additional Quantitative Results

## D.1 Details on Video-to-Audio Comparison

For the results in Table 1, we reproduce the results of Seeing and Hearing (Xing et al., 2024), V-AURA (Viertola et al., 2024), FoleyCrafter (Zhang et al., 2024), and MMAudio (Cheng et al., 2024a) using the official code and pre-trained models. For the other baselines, we use the generated samples provided by the authors, i.e., Frieren, V2A-Mapper, and Movie Gen (Polyak et al., 2024). Furthermore, the $CLAP_{cap}$ scores of MMAudio, MovieGen, and ThinkSound are 0.43, 0.44, and 0.49, respectively.

## D.2 Impact of Model Size

We compare three model size of ThinkSound: **Large (1.3B)**, **Medium (724M)**, and **Small (533M)**. The Large model achieves the best performance across all metrics as shown in Table 9. These results demonstrate that model capacity significantly enhances audio quality and improves alignment with ground truth distribution. As model size decreases, performance degrades substantially, highlighting the necessity of adequate model capacity for effective audio generation.

Table 9: Impact of model size results.

| Size | FD↓ | $KL_{PaSST}$ ↓ | $KL_{PaNNs}$ ↓ | DeSync↓ | $CLAP_{CoT}$ ↑ |
|---|---|---|---|---|---|
| Small | 43.26 | 1.64 | 1.39 | 0.50 | 0.43 |
| Medium | 37.62 | 1.56 | 1.34 | 0.47 | 0.44 |
| Large | **34.56** | **1.52** | **1.32** | **0.46** | **0.46** |

## D.3 Performance across different difficulty levels

To better validate the performance of our CoT-Guided generation, we also report the results in the video-to-audio generation of different difficulty levels. We illustrate the construction of different difficulty levels in Section A A.3. The results are shown in Table 10, and we can conclude that (1) As expected, the performance of all models decreases as the difficulty level increases, and (2) Our CoT-Guided generation outperforms other baselines across all difficulty levels.

Table 10: Performance across different difficulty levels.

| Difficulty | FD↓ | $KL_{PaSST}$ ↓ | $KL_{PaNNs}$ ↓ | DeSync↓ | $CLAP_{CoT}$ ↑ |
|---|---|---|---|---|---|
| Easy | **31.32** | **1.35** | **1.16** | **0.42** | **0.52** |
| Medium | 35.45 | 1.46 | 1.31 | 0.46 | 0.49 |
| Hard | 48.78 | 1.63 | 1.40 | 0.57 | 0.41 |

## D.4 Performance Comparisons between coarse-grained and fine-grained CoT

To further validate the effectiveness of our fine-grained CoT, we compare the performance of our model with the coarse-grained CoT. The results are shown in Table 11. We can conclude that our fine-grained CoT outperforms the coarse-grained CoT across all metrics.

Table 11: Performance comparisons between coarse-grained and fine-grained CoT.

| Granularity | FD↓ | $KL_{PaSST}$ ↓ | $KL_{PaNNs}$ ↓ | DeSync↓ | $CLAP_{CoT}$ ↑ |
|---|---|---|---|---|---|
| Coarse | 42.72 | 1.58 | 1.41 | 0.52 | 0.34 |
| Fine | **34.56** | **1.52** | **1.32** | **0.46** | **0.46** |

## D.5 Effectiveness of T5 Encoder for CoT Structure

A key assumption of our work is that the T5 encoder can effectively capture the logical structure within the CoT reasoning steps. We provide both a theoretical rationale and empirical validation for this.

**Theoretical Rationale**    While T5 is pre-trained on general web text (C4), its Transformer architecture is fundamentally designed to capture long-range dependencies and compositional structure. The reasoning steps in our CoT (e.g., "first do A, then B happens, which causes C") are expressed through natural language syntax and logical connectives. T5's contextual embeddings are well-suited to capture these structural and causal cues. The goal is not for T5 to "understand" reasoning in a human sense, but to produce a rich, structured conditioning signal that the downstream diffusion transformer can effectively leverage.

**Empirical Validation**    To prove that the T5 encoder effectively utilizes the structure of the CoT, we conducted an ablation study comparing our full model against two degraded variants: (1) **Tags Only**, which provides the model with only extracted keywords (e.g., "car," "rain," "dog bark"), removing all reasoning structure; and (2) **Randomized CoT**, which randomly shuffles the sentences within the CoT, preserving all keywords but destroying the logical flow. As shown in Table 12, both ablations lead to a significant performance drop. This strongly indicates that the performance gain is attributable to the stepwise logical structure of the CoT—as encoded by T5—and not merely the presence of more keywords.

Table 12: Ablation study on the importance of CoT structure. We compare our full model against variants with only keywords (Tags Only) and shuffled reasoning sentences (Randomized CoT). Results show that the logical structure is critical for performance.

| Setting | FD$\downarrow$ | KL$_{\text{PaSST}} \downarrow$ | KL$_{\text{PaNNs}} \downarrow$ | DeSync$\downarrow$ | CLAP$_{\text{CoT}} \uparrow$ |
|---|---|---|---|---|---|
| Randomized CoT | 40.52 | 1.56 | 1.35 | 0.51 | 0.43 |
| Tags Only | 39.35 | 1.60 | 1.38 | 0.50 | 0.42 |
| Ours (Ordered CoT) | **34.56** | **1.52** | **1.32** | **0.46** | **0.46** |

## D.6    Impact of CoT Verbosity on Generation Quality

To mitigate the risk of verbose or hallucinated CoT text, we employ a two-pronged strategy: (1) rigorous quality control during the creation of our AudioCoT training data (see Appendix A.2), and (2) direct constraints on CoT generation during inference (e.g., $\leq 3$ sentences, $\leq 77$ tokens). To empirically validate this approach, we conducted an ablation study quantifying the impact of verbosity. We removed our length constraints to generate "Overly Detailed" CoT prompts and compared them against our proposed "Concise CoT." The results in Table 13 demonstrate a clear performance degradation with verbose reasoning, confirming that our dual strategy is effective and that concise reasoning is critical for achieving high-quality results.

Table 13: Impact of CoT verbosity. We compare our standard concise CoT against an overly detailed version generated without length constraints. Verbose reasoning leads to a clear performance degradation.

| Setting | FD$\downarrow$ | KL$_{\text{PaSST}} \downarrow$ | KL$_{\text{PaNNs}} \downarrow$ | DeSync$\downarrow$ | CLAP$_{\text{CoT}} \uparrow$ |
|---|---|---|---|---|---|
| Overly-detailed CoT | 43.56 | 1.61 | 1.55 | 0.54 | 0.35 |
| Concise CoT (Ours) | **34.56** | **1.52** | **1.32** | **0.46** | **0.46** |

## D.7    Dedicated MLLM Reasoning Evaluation

The quality of the CoT reasoning is the linchpin of our framework. To substantiate our claims, we conducted a comprehensive evaluation of the generated CoT quality using both expert human annotators and automated LLM-based metrics.

**Human Evaluation**    We randomly sampled 100 VGGSound test cases. Two expert annotators rated each generated CoT on a 0-1 scale across five dimensions: (1) multimodal integration, (2) specificity of audio details, (3) feasibility for audio generation, (4) logical consistency, and (5) brevity/format compliance. The final score is the sum (max 5).

**LLM-Based Similarity Evaluation** For large-scale analysis, we compared the generated CoTs to ground-truth CoTs using GPT-4-nano. The model was prompted to score similarity on a 0–5 scale, focusing on reasoning structure, causal/temporal relationships, and object-sound associations.

**Results** The results, presented in Table 14, provide strong empirical evidence that our fine-tuned MLLM (ThinkSound) greatly outperforms other powerful MLLMs in generating high-quality, structured CoT for video-to-audio reasoning.

Table 14: Evaluation of MLLM-generated CoT quality. Our model is compared against baselines using human evaluation and LLM-based similarity scoring, demonstrating superior performance in generating high-quality CoT.

| Model | Human Score (0-5) | LLM CoT Similarity (0-5) |
|---|---|---|
| Qwen2.5-VL-7B | 3.78 | 3.95 |
| Qwen2-Audio-7B | 3.82 | 4.09 |
| ThinkSound (VideoLLaMA2) | **4.13** | **4.31** |

# E   Limitation and Future

While the current MLLMs are capable of a strong understanding and reasoning of semantic information, they still have limitations in understanding the precise temporal and spatial information of video. For example, in the case of locating the exact timestamp of the sound event, MLLMs often fail to provide accurate results or provide wrong results. Moreover, the current open-source video-audio datasets for audio generation are limited in diversity and coverage, which may lack rare or culturally specific sound events. In the future, we will continue to explore more diverse and comprehensive datasets to improve the performance of our model. Furthermore, we will explore more effective methods to improve the temporal and spatial alignment of generated audio.

# F   Potential Negative Societal Impacts

ThinkSound carries potential risks if misused. Malicious actors could exploit the system to generate fake audio for synthetic media, thereby contributing to the spread of misinformation. Moreover, if the training data underrepresents certain cultures or environments, the model may unintentionally amplify biases—for instance, by reinforcing stereotypes or misassociating sounds with particular demographic groups.

## F.1   Ethical Considerations

**The dataset used in this research is strictly for academic and non-commercial purposes.** We implemented several measures to ensure compliance with ethical standards, as follows.

- **Data Transparency and Anonymization.** We only provide ASR transcripts after rigorous text anonymization processes, visual features of video clips, our annotations, and links to the original videos, to ensure transparency regarding the data sources and their usage while maintaining anonymity.

- **Authorization.** Any personal data should be used only with express authorization, ensuring lawful and fair processing in accordance with applicable laws.

# G   Safeguards

We used a diverse training dataset covering a wide range of acoustic scenes to minimize reinforcing stereotypes or incorrect associations between sounds and specific demographic groups. The model will be released in stages to better assess its impact and improve safeguards. However, once the model is openly released, we cannot control how others use it. Therefore, we provide clear usage guidelines to encourage responsible use and help mitigate potential misuse.

