# OpenReview forum: "ThinkSound: Chain-of-Thought Reasoning in Multimodal LLMs for Audio Generation and Editing"
_NeurIPS.cc/2025/Conference — NeurIPS 2025 poster_

### Official Review · Reviewer_MPd8 · 2025-06-25

**Clarity:** 2
**Significance:** 3
**Originality:** 2
**Rating:** 4
**Confidence:** 3

**Summary:**

This paper proposes ThinkSound, a novel framework for interactive audio generation and editing for videos wit CoT reasoning. The methods includes three stages, with foundational foley generation, interactive object-centric refinement, and targeted editing guided by instructions. Besides, a dataset with structural reasoning annotations AudioCoT is proposed. Experiments show the capabilities in V2A generation in audio metrics, CoT metrics, and Movie Gen Audio benchmark.

**Questions:**

1. L155-160, how is instruction-based audio constructed? How to get paired audio 1 and audio 2? Is it a random process or meticulous one?

2. Why use CoT in audio generation? Better to provide qualitative comparison of w/wo CoT process (for example, in Fig.2, what will the integrated analysis be if wo CoT)

3. Why adopt such an architecture of fig. 3 right part? Can the design of the MM-DiT be validated effectively? (e.g. the ablation study of adaptive fusion)

**Ethical Concerns:**

["NO or VERY MINOR ethics concerns only"]

**Final Justification:**

After reviewing the detailed response from the authors, I find that my initial concerns have been addressed. However, I still believe that the qualitative comparison between the method with and without the CoT process, which is not included in the current submission and cannot be incorporated in the rebuttal, needs further validation to demonstrate the full effectiveness of the method.

Additionally, I share the concerns raised by Reviewer 2uba regarding the robustness of the method. There appear to be certain limitations and flaws that may affect its generalizability or practical application. Given these issues, I am still on the borderline about my recommendation.

**Limitations:**

Yes.

**Quality:**

2

**Strengths And Weaknesses:**

Strength:

1.The paper proposes a novel three-stage interactive framework for V2A for soundscapes through CoT reasoning.

2.A large-scale multimodal dataset AudioCoT with detailed CoT annotation for visual, textual, and audio description is proposed.

3.The experiments validate the proposed method’s performance in audio generation.


Weakness (details see “Questions”):

1.The clarity of dataset construction and training architecture needs to improve.

2.The effectiveness and necessity of incorporating CoT reasoning need to be validated qualitatively

3.The design module of MM-DiT architecture needs more experiments to validate.

---

> ### Author Rebuttal · Authors · 2025-07-30
>
> ## Response to Reviewer MPd8
>
> We thank you for the reviewer's constructive feedback and for `recognizing the novelty of our three-stage framework` and `the contribution of the AudioCoT dataset`. We hope the following point-by-point response will fully address your concerns. We are committed to open-sourcing all code, data, and models upon acceptance.
>
> ---
>
> > **1. On the Clarity of the Data Construction Pipeline for Audio Editing**
>
> Thank you for this critical question. We wish to emphasize that our editing dataset is constructed through a **meticulous, multi-stage pipeline, not a random process**. This systematic approach, detailed in **Supplementary A.2**, is designed to produce high-quality, semantically coherent, and contextually plausible training data.
>
> Here is a step-by-step breakdown of our process for creating each `(instruction, input_audio, target_audio)` triplet:
>
> 1.  **Semantic Pairing and Filtering:** We begin by programmatically identifying audio components for editing (e.g., for an addition or removal task). To ensure the resulting instruction is unambiguous, we perform semantic filtering based on audio event tags. For example, we exclude pairs with overlapping sounds (e.g., avoiding a "remove engine" instruction in a scene also tagged with "passing car"). We then score potential pairs for contextual plausibility, ensuring that combined sounds are logical within the scene's context.
>
> 2.  **Instruction-CoT Generation:** Once a valid pair of audio components is identified, we use GPT-4 to generate a natural language instruction and its corresponding CoT. This instruction is based on the desired editing operation and is grounded in the video's content by leveraging the original CoT from Stage 1.
>
> 3.  **Paired Audio Synthesis:** We then programmatically synthesize the input_audio and arget_audio. For an "addition" task, we might combine two clips (birdsong + rain) to create the target_audio, while the original birdsong clip serves as the input_audio. This creates a perfectly aligned training pair for the editing model.
>
> 4.  **Rigorous Human Verification:** To guarantee quality, we implement a **human-in-the-loop verification protocol**. A random 5% sample of the generated data is manually reviewed at each stage. If the human rejection rate exceeds a **5% threshold**, we recalibrate our automated filters and re-process the entire batch. This iterative feedback loop ensures the perceptual quality and semantic integrity of our final training data.
>
> We will integrate this more detailed description directly into the main body of our paper (Section 3.2) to fully articulate the rigor of our data generation methodology.
>
> ---
>
> > **2. On the Qualitative Necessity and Impact of CoT Reasoning**
>
> Thank you for your feedback. While our quantitative results in Tables 1, 3, 4, and 5 consistently show that CoT provides significant gains, qualitative evidence is helpful to understanding *why*.
>
> CoT enables our model to move beyond generating a generic "sound blob" and instead synthesize a **structured soundscape with correct temporal and causal relationships**. For example, given a video of a person walking to a door and knocking:
>
> *   **Without CoT**, a model might generate a temporally confused mixture of "footsteps" and "knocking" sounds, potentially out of sync or overlapping incorrectly.
> *   **With CoT**, the MLLM generates a plan like: "_First, there is the sound of footsteps on a wooden floor, which grow louder. Then, after the footsteps stop, there is a sharp knocking sound on the door._" This explicit plan guides the audio model to generate the events in the correct sequence and with the correct acoustic properties.
>
> We fully agree that a visual comparison is the best way to show this. **We will add a new qualitative comparison figure to the appendix.** Similar to Figure 4, it will show side-by-side spectrograms of audio generated from the same video clip, with and without CoT, directly illustrating how CoT enables superior temporal structure and event accuracy.
>
> ---
>
> > **3. On the Design and Validation of the MM-DiT Architecture**
>
> Thank you for your suggestion. Our enhanced MM-DiT is specifically designed to effectively process and fuse the complex, multimodal streams inherent to our task. We validate its design with both existing ablations from the paper and a new targeted experiment.
>
> **Validation of Key Components:** We provide three key pieces of empirical evidence:
>
> 1.  **Ablation on Gated Fusion (Existing):** As the reviewer noted, we investigated the importance of our adaptive fusion module. **Table 6 in our paper already provides this ablation.** It shows that our gated fusion mechanism greatly outperforms simpler linear fusion or using audio features alone, confirming the value of this specific design choice.
>
> 2.  **Ablation on Dual Text-Encoder Strategy (Existing):** We validated our choice to use two complementary text encoders. As shown in **Table 5 of our paper**, using both CLIP-encoded captions (for global context) and T5-encoded CoT (for CoT texts) outperforms using either one alone.This result confirms that both conditioning signals are necessary and complementary for achieving the best performance.
>
> 3.  **Ablation on the Full MM-DiT Architecture (New):** To validate the overall multimodal architecture against a simpler baseline, we conducted a new experiment comparing our full MM-DiT to a standard DiT model conditioned only on text (CoT) and global CLIP embeddings, without the multi-stream transformer blocks.
>
>
> | **Method** | FD↓ | **$\text{KL}_{\text{PaSST}}$**↓ | **$\text{KL}_{\text{PaNNs}}$**↓ | DeSync↓ | **$\text{CLAP}_{\text{CoT}}$**↑ |
> | --- | --- | --- | --- | --- | --- |
> | DiT | 45.20 | 1.64 | 1.45 | 0.56 | 0.39 |
> | MM-DiT | **34.56** | **1.52** | **1.32** | **0.46** | **0.46** |
>
> The results show a dramatic improvement across all metrics, which confirms that the multi-stream design is crucial for multimodal audio generation.

---

> > ### Author Response · Authors · 2025-08-05
> >
> > We are deeply grateful for your invaluable review, and we sincerely appreciate the considerable time and expertise you have devoted to evaluating our work. We have endeavored to address all your concerns thoroughly in our point-by-point rebuttal and would be most grateful for any further discussion that address any remaining questions about our work.

---

> > > ### Comment · Reviewer_MPd8 · 2025-08-06
> > >
> > > Thank you to the authors for addressing most of my concerns. However, I noticed that in Supplementary A.2, the BBC Sound Effects are cited, but the jump link for further explanation seems to be broken. I have a few concerns regarding the ethics and licensing:
> > >
> > > Ethical Considerations: Are there any ethical issues related to the use of the BBC Sound Effects, especially in the context of their origin and usage in the dataset?
> > >
> > > Licensing for Academic Use: Is the BBC license appropriate for academic use as stated in Supplementary F.1, and are the terms clearly specified? Given that the dataset appears to be used in a research context, it would be helpful to confirm that it complies with the appropriate usage rights.
> > >
> > > Open-sourcing of Dataset: Will the dataset, including the BBC Sound Effects, be open-sourced, and if so, under what conditions?

---

> > > > ### Author Response · Authors · 2025-08-06
> > > > **Follow-Up Response**
> > > >
> > > > Thank you very much for your valuable review and for bringing up these important questions regarding the use of BBC Sound Effects data, as well as the associated ethical and licensing considerations. We truly appreciate your attention to these details. Please find our detailed responses below:
> > > >
> > > > ---
> > > >
> > > > **1. Broken Link in Supplementary Material**
> > > > We apologize for the broken jump link in the supplementary. **The correct link to the official BBC Sound Effects website is provided at the end of Section 3.1 in the main text.** For all audio-only datasets, including BBC Sound Effects, Freesound, and AudioSet-SL, our downloads are based on WavCaps[1], a widely used, fully open audio dataset for research purposes. Given its open and research-oriented nature, we believe there are no ethical concerns regarding our data usage.
> > > >
> > > > ---
> > > >
> > > > **2. BBC License for Academic Use**
> > > > We have reviewed the official BBC license and confirm that it permits academic and non-commercial use, which aligns with our usage as described in Supplementary F.1.
> > > > As stated in the license:
> > > > > Provided you keep to these rules, the BBC grants you permission to use the BBC content but only…
> > > > > For non-commercial, personal or **research purposes** (for example, including the content on a non-commercial, advertisement-free reminiscence website aimed at helping trigger memories in people with dementia)
> > > > > For formal education purposes while you are a student or a member of staff of a school, college or university (for example, if you are enrolled on a university or college course, or if you are a school pupil, or you are a teacher and you wish to display the content on an electronic whiteboard, including images in a printed class worksheet)
> > > >
> > > > Therefore, our use of the BBC Sound Effects strictly complies with these terms.
> > > >
> > > > ---
> > > >
> > > > **3. Open-sourcing of the Dataset**
> > > > We will release our AudioCoT dataset on HuggingFace, which will include metadata entries such as dataset name, id, url, caption, and CoT. We **will not** release raw data (such as audio or video files) from the original datasets. Instead, we will recommend users to download the original audio data directly from the official sources such as WavCaps and VGGSound. Clear licensing information and usage restrictions will be included in our open repository to ensure compliance.
> > > >
> > > > ---
> > > >
> > > > Thank you again for your careful review and attention to these important issues. We will make these clarifications explicit in our revised manuscript.
> > > >
> > > > ---
> > > >
> > > > **[Reference]**
> > > >
> > > > [1] Mei, Xinhao, et al. "Wavcaps: A chatgpt-assisted weakly-labelled audio captioning dataset for audio-language multimodal research." IEEE/ACM Transactions on Audio, Speech, and Language Processing 32 (2024): 3339-3354.

---

> > > > > ### Comment · Reviewer_MPd8 · 2025-08-06
> > > > >
> > > > > Thanks to the author for the detailed response. My concerns are well addressed.

---

> > > > > > ### Author Response · Authors · 2025-08-06
> > > > > > **Gratitude for Reviewer Feedback and Score Consideration**
> > > > > >
> > > > > > Thank you very much for your positive response. **We are glad that all your concerns have been well addressed.** We are truly pleased to hear this, and we sincerely appreciate the time and effort you dedicated to reviewing our paper and providing such helpful suggestions.
> > > > > >
> > > > > > **If you feel that our revisions have satisfactorily resolved the concerns, we would sincerely appreciate your consideration for a possible score increase.** Thank you again for your thoughtful review and kind support.

---

### Official Review · Reviewer_2uba · 2025-07-02

**Clarity:** 3
**Significance:** 2
**Originality:** 3
**Rating:** 4
**Confidence:** 4

**Summary:**

The paper introduces ThinkSound, a three-stage framework that brings chain-of-thought (CoT) reasoning into video-to-audio generation. A fine-tuned VideoLLaMA 2 model first produces explicit, step-by-step CoT instructions; these then guide a single, multimodal flow-matching audio foundation model that can (1) create an initial “foley” soundtrack, (2) refine sounds for user-clicked regions, and (3) perform text-guided edits such as adding or removing events. To train and evaluate this pipeline, the authors curate AudioCoT, a large dataset pairing videos, audio/text clips, and structured CoT annotations. Experiments on VGGSound show clear gains over recent V2A baselines (e.g., MMAudio) in fidelity, semantic alignment, and human MOS, while an out-of-distribution test on the MovieGen Audio Bench confirms strong generalisation. Ablations reveal large drops when CoT reasoning or the gated video-audio fusion is removed, underscoring the importance of both components.

**Questions:**

My questions are listed in Strengths And Weaknesses. Specifically:

- I may have missed where the editing training data is coming from. Can the authors highlight that part?
- Have the authors tried verifying how clean or noisy the generated data is? Any data verification step?
- Stage 2: Interactive Object-Focused Audio Generation is supposed to be click-based and done by humans -- how did the authors manage to do an objective evaluation? What did I miss?

**Ethical Concerns:**

["NO or VERY MINOR ethics concerns only"]

**Final Justification:**

The authors have made efforts. The method has some flaws and not very robust. I am borderline.

**Limitations:**

Yes

**Quality:**

2

**Strengths And Weaknesses:**

### Strengths:

- The paper is nicely written and well presented.
- The paper tries to solve a relevant problem of video-audio generation, which is hot now.
- The paper is well structured and thought out.
- The results look promising.

### Weaknesses:

- The title is a bit misleading. The title made me think that somehow the authors propose CoT in the audio generation process (also the paper is video-to-audio and not just audio, which should also be highlighted) -- which ideally means that the model thinks step by step to generate audio. However, the paper generates thinking in text, which is input to a DiT-like model by conditioning -- so ideally, the generation process is not what is thinking. The step-by-step in the process also comes through additional steps in generation (which are kind of optional) -- but thinking steps are not happening while generating audio -- which is what CoT ideally is.
- ⁠The Cost is just conditioned to the DiT with a T5. I am doubtful if any model (DiT in this paper) would understand thought processes through T5 embeddings. T5 embeddings, in prior literature, is good at passing information about objects, attributes, etc -- the fact that it can pass knowledge about thought processes is something I am not being able to comprehend (maybe some ablation is needed to justify this) -- and it is important as the paper is based on this.
- ⁠Figure 3 left is a bit misleading in the sense that the DiT should not be on top of the Decoder -- this ideally indicates a different form of architecture -- the DiT gets the output of MLLM in a cascaded fashion, and it's not joint.
- About Section 4.4:
  - Object-focused audio Generation is not new. See [this paper](https://arxiv.org/abs/2506.04214) for example.
  - I may have missed where the editing training data is coming from. Can the authors highlight that part?
- About Section 3.2
  - Have the authors tried verifying how clean or noisy the generated data is? Any data verification step?

---

> ### Author Rebuttal · Authors · 2025-07-30
>
> ## Response to Reviewer 2uba
>
> We thank the reviewer for the constructive feedback, as well as for recognizing our work as `nicely written and well presented`, `well structured and thought out`, and with `promising results`. We are fully committed to open-sourcing all code, data, and benchmarks upon acceptance. We hope the following point-by-point responses will completely address all remaining concerns.
>
> ---
>
> > **1. On the Title and the Nature of "Thinking"**
>
> Thank you for the opportunity to clarify our definition and the specific role of "Chain-of-Thought" (CoT) in our framework, particularly regarding the title.
>
> *   **Our Contribution:** The core innovation of ThinkSound is **not** to embed a step-by-step generative process within the audio synthesis model itself. Rather, as described in the **Abstract (L10-12)**, our contribution is the introduction of an **explicit, interpretable, and controllable reasoning stage**—powered by an MLLM—that ***precedes and guides*** the audio foundation model. We expect this would be similar to the text-form reasoning LLM, which begins with a pure thinking process and then generates the whole final response. This MLLM-generated textual CoT serves as a detailed blueprint, decomposing the complex video-to-audio task into manageable sub-problems, much like a planning module.
>
> *   **Motivation and Analogy:** Our approach is deliberately designed to mimic the workflow of professional sound designers. They first analyze a visual scene, reason about the sequence of events and acoustic properties, and *only then* proceed to synthesize and layer sounds. Existing end-to-end models bypass this critical reasoning step, treating generation as an opaque transformation. By making the reasoning explicit, ThinkSound gains significant advantages in interpretability, controllability, and fidelity, as our results demonstrate.
>
> *   **Consistency with an Emerging Paradigm:** This use of CoT—where a large model generates a textual reasoning plan to steer a specialized downstream model—is a powerful and emerging paradigm in multimodal AI, as acknowledged by Reviewer Ajam and 2MyN. It is methodologically consistent with recent works such as **ImageGen-CoT [1], DeepSound-V1 [2], and MINT [3]**, which also leverage LLM-generated text-based reasoning to guide complex generative tasks.
>
> We will revise the introduction and methodology sections to make this architectural choice and our motivation crystal clear.
>
> ---
>
> > **2. On the Use of T5 Embeddings for Encoding Reasoning**
>
> Thank you for this insightful question. We address it with both a theoretical rationale and targeted empirical evidence.
>
> *   **Theoretical Rationale:** While T5 is pre-trained on general web text (C4), a massive English-language dataset containing approximately 750GB of clean web text, its Transformer architecture with pre-training paradigm is fundamentally designed to capture long-range dependencies and compositional structure within sequential data. The reasoning steps in our CoT (like "first do A, then B happens, which causes C") are expressed through natural language syntax and logical connectives. **T5's contextual embeddings are suitable to capture these structural and causal cues.** Furthermore, the goal is not for T5 to "understand" reasoning in a human sense, but to **produce a rich, structured conditioning signal** that the downstream diffusion transformer can effectively leverage.
>
> *   **Empirical Validation:** To prove that the T5 encoder effectively utilizes the *structure* of the CoT, we conducted an ablation study to isolate the contribution of logical structure versus mere token content. We compared our full model against two degraded variants:
>     1.  `Tags Only`: Provided the model with only extracted keywords (e.g., "car," "rain," "dog bark"), removing all reasoning structure.
>     2.  `Randomized CoT`: Randomly shuffled the sentences within the CoT, preserving all keywords and sentence content but destroying the logical flow.
>
> The results unequivocally show that the ordered, logical structure is critical for performance:
>
> | **Setting**             | FD↓ | **$\text{KL}_{\text{PaSST}}$**↓ | **$\text{KL}_{\text{PaNNs}}$**↓ | DeSync↓ | **$\text{CLAP}_{\text{CoT}}$**↑ |
> | ----------------------- | ---------------- | ---------------- | ---------------- | ---------------- | ---------------- |
> | Randomized CoT          | 40.52            | 1.56             | 1.35             | 0.51             | 0.43             |
> | Tags Only               | 39.35            | 1.60             | 1.38             | 0.50             | 0.42             |
> | **Ours (Ordered CoT)**  | **34.56**        | **1.52**         | **1.32**         | **0.46**         | **0.46**         |
>
> As shown, both ablations lead to a significant performance drop across key metrics. This strongly indicates that the performance gain is attributable to the **stepwise logical structure of the CoT—as encoded by T5—and not merely the presence of more keywords or longer context.**
>
> ---
>
> > **3. On the Architectural Clarity of Figure 3**
>
> We thank the reviewer for this feedback. The reviewer's understanding of our architecture as a **cascaded pipeline** is perfectly correct, as we describe in the manuscript (e.g., Sec. 4.1, L167-168). We agree that the diagram could better reflect this flow, and we will revise Figure 3 in the final version to be more explicit and self-contained.
>
> ---
>
> > **4. On the Novelty of Object-Focused Generation**
>
> Thank you for bringing this concurrent work (`arXiv:2406.04214`) to our attention. As this paper appeared on arXiv after the NeurIPS submission deadline, we were unaware of it. We will gladly cite and discuss it in our related work section. However, our work remains clearly distinct in several critical aspects:
>
> *   **Input Modality:** Our method operates on **videos**, not static images. This requires complex spatio-temporal reasoning about events and motion that is fundamentally absent in image-based generation.
> *   **Framework Scope:** Our object-focused stage is one component of a comprehensive, three-stage interactive pipeline that also includes foundational foley generation and instruction-based editing, all unified under a single model architecture.
> *   **Core Method:** Our primary novelty is the **explicit CoT reasoning** that guides the entire generation process, a fundamentally different mechanism from the cited work.
>
> ---
>
> > **5. On Data and Evaluation Details**
>
> We apologize for any lack of clarity and provide the requested details below, with pointers to the supplementary.
>
> *   **(a) Editing Data Source:** The training data for audio editing was programmatically generated from our AudioCoT dataset, as detailed in **Supplementary A.2 (Stage 3)**. Specifically, we used a powerful LLM (GPT-4) to semantically select and pair source/target audio segments from our dataset. The LLM then generated a corresponding natural language instruction (e.g., *"Add the sound of rain to the existing birdsong"*). This automated procedure creates high-quality and contextually plausible `(instruction, input_audio, output_audio)` triplets, which are essential for training a robust editing model.
>
> *   **(b) Data Quality Verification:** This is a crucial point. We implemented a rigorous, multi-step quality control pipeline for AudioCoT, detailed in **Supplementary A.2 (Quality Control)**. This process includes:
>      1. **Automated Filtering:** We use CLAP scores to automatically identify and discard audio-CoT pairs with poor semantic alignment, ensuring the MLLM learns from relevant examples.
>      2. **LLM-based Validation:** We leverage GPT-4.1-nano as a validation agent to check the semantic plausibility and logical consistency of generated editing instructions, preventing illogical or "hallucinated" steps from entering our dataset.
>     3. **Human Verification:** At each stage, a 5% random sample of the data is manually reviewed by our team. If the rejection rate exceeds a 5% threshold, we recalibrate our automated filters and reprocess the entire batch.
>
>     This process ensures the high quality of our dataset. Furthermore, as shown in our **response to Reviewer 2MyN (Q2)**, we also objectively and subjectively evaluated the CoT data generated by our fine-tuned MLLM, confirming it is high-fidelity and suitable for downstream conditioning.
>
> *   **(c) Interactive Stage Evaluation:** We thank the reviewer for this feedback. For our quantitative experiments (Table 3), we adopted a standard and reproducible methodology: **simulating user interaction**. As detailed in **Supplementary A.3**, we used ground-truth object bounding boxes from the videos to serve as automated "user clicks." This allowed us to generate audio for a targeted region and then objectively compare it against the corresponding ground-truth audio for that same region using metrics like FD and KL. This approach provides a repeatable and fair benchmark for the interactive generation capability.
>
> ---
>
> ### **References**
>
> [1] Liao, J., et al. "Imagegen-cot: Enhancing text-to-image in-context learning with chain-of-thought reasoning." arXiv preprint arXiv:2403.19312 (2024).
>
> [2] Liang, Y., et al. "DeepSound-V1: Start to Think Step-by-Step in the Audio Generation from Videos." arXiv preprint arXiv:2403.22208 (2024).
>
> [3] Wang, Y., et al. "Mint: Multi-modal chain of thought in unified generative models for enhanced image generation." arXiv preprint arXiv:2403.01298 (2024).

---

> > ### Author Response · Authors · 2025-08-05
> >
> > We are deeply grateful for your invaluable review, and we sincerely appreciate the considerable time and expertise you have devoted to evaluating our work. We have endeavored to address all your concerns thoroughly in our point-by-point rebuttal and would be most grateful for any further discussion that address any remaining questions about our work.

---

> > ### Comment · Reviewer_2uba · 2025-08-05
> > **Reply**
> >
> > The T5 analysis shown is good but there are a lot of papers which show CLAP is similar to BoW [1,2]. Thus using it to condition may not be a great choice.
> >
> > Similar to other reviewers, I agree that the data construction is not very clear in the paper. I checked Supplementary A.2 (Stage 3) -- looks like pairs of original audios were used -- in this case, how does editing make sense given the components you are not editing in the instruction may sound different (as they are recorded in a different setting).
> >
> > The problem tacked is solid. The effort in paper and rebuttal solid. I only do not have full confidence on the robustness of the approach and find some flaws. Given this, I am raising my score to 4.
> >
> > Thank You and good luck!

---

> ### Author Response · Authors · 2025-08-06
> **Follow-Up Response**
>
> Thank you sincerely for your positive adjustment of the score and for your continued constructive feedback. We greatly appreciate your recognition that `the problem tacked is solid` and `the effort in paper and rebuttal solid`. We hope the following point-by-point explanations will fully address your remaining concerns.
>
> ---
>
> **1. Clarification on Text Conditioning Strategy**
>
> Thank you for raising this important point about text conditioning. We want to ensure we fully understand your concern. Are you referring to our use of CLAP as a text encoder for conditioning the audio generation model? If so, we would like to respectfully clarify our actual text encoding approach:
>
> - **We do not use CLAP as a conditioning mechanism for our audio generation model.** Rather, CLAP is used exclusively as an objective evaluation metric for measuring audio-text alignment quality during evaluation.
>
> - Our actual text encoding strategy consists of:
>     - **CLIP Encoder:** Used to encode the original VGGSound captions, which are typically simple audio event tags (e.g., "dog barking," "rain"). This provides global guidance for basic audio event information.
>      - **T5 Encoder:** Used specifically to encode our CoT text, leveraging its strong contextual understanding capabilities.
>
> - **This dual-encoder strategy enables us to incorporate both simple event-level information (via CLIP) and complex reasoning structure (via T5), providing comprehensive conditioning for our audio generation model.**
>
> If your concern relates to a different aspect of our text conditioning approach, we would greatly appreciate your clarification so that we can address it more precisely.
>
> ---
>
> **2. Clarification on Audio Editing Data Construction**
>
> We sincerely apologize for not providing clearer details in the main paper, and we commit to significantly improving Section 3.2 in the revised version to make our data construction methodology fully transparent.
>
> To address the scarcity of instruction-based editing datasets, we designed a synthetic data construction strategy that simulates editing operations such as addition and removal, while maintaining data quality through both automated and manual checks. Our process is as follows:
>
> 1. **Semantic Pairing and Filtering:** We begin by programmatically identifying audio components for editing (e.g., for an addition or removal task). To ensure the resulting instruction is unambiguous, we perform semantic filtering based on audio event tags. For example, we exclude pairs with overlapping sounds (e.g., avoiding a "remove engine" instruction in a scene also tagged with "passing car"). We then score potential pairs for contextual plausibility, ensuring that combined sounds are logical within the scene's context.
>
> 2. **Instruction-CoT Generation:** Once a valid pair of audio components is identified, we use GPT-4 to generate a natural language instruction and its corresponding CoT. This instruction is based on the desired editing operation and is grounded in the video's content by leveraging the original CoT from Stage 1.
>
> 3. **Paired Audio Synthesis:** We then programmatically synthesize the input_audio and arget_audio. For an "addition" task, we might combine two clips (birdsong + rain) to create the target_audio, while the original birdsong clip serves as the input_audio. This creates a perfectly aligned training pair for the editing model.
>
> 4. **Rigorous Human Verification:**
> To guarantee quality, we implement a **human-in-the-loop verification protocol**. A random 5% sample of the generated data is manually reviewed at each stage. If the human rejection rate exceeds a **5% threshold**, we recalibrate our automated filters and re-process the entire batch. This iterative feedback loop ensures the perceptual quality and semantic integrity of our final training data.
>
> ---
>
> Thank you again for your thoughtful review and valuable insights.

---

> ### Author Response · Authors · 2025-08-08
> **With Appreciation: Kindly Requesting Further Feedback**
>
> We would like to sincerely thank you once again for your thoughtful review and for increasing your score. **We are deeply grateful for your recognition of the solid problem tackled and the efforts made in both the paper and the rebuttal.** Your constructive feedback has been invaluable in refining our work.
>
> **In our previous follow-up, we have endeavored to address all of your remaining concerns as thoroughly as possible. If there are any aspects that still require clarification, or if you have any additional questions, please feel free to let us know. We sincerely value your expert perspective and would be more than happy to provide any further information or clarification.**
>
> Thank you once again for your time, constructive guidance, and support throughout the review process.

---

### Official Review · Reviewer_2MyN · 2025-07-02

**Clarity:** 3
**Significance:** 3
**Originality:** 2
**Rating:** 4
**Confidence:** 4

**Summary:**

This paper introduces ​ThinkSound, a novel framework for video-to-audio (V2A) generation that leverages ​Chain-of-Thought (CoT) reasoning​ to address the limitations of existing end-to-end approaches in capturing nuanced audio-visual relationships. Key innovations include ​AudioCoT—the first dataset with structured audio reasoning annotations —and a ​unified audio foundation model​ using flow matching with gated multimodal fusion.

**Questions:**

The reported inference speed is impressive, particularly its advantage over MMAudio. However, this result seems somewhat counter-intuitive, as the ThinkSound pipeline involves two distinct models: an MLLM for CoT generation and the flow matching model for audio synthesis. Could the authors provide a more detailed breakdown of this latency measurement? Specifically:
1. How was the total inference time calculated? Does it include the MLLM's generation time?
2. What are the primary factors contributing to this high efficiency? A more in-depth analysis explaining why the proposed framework is significantly faster than a diffusion-based model like MMAudio, despite its two-component architecture, would be highly beneficial for the reader.

**Ethical Concerns:**

["NO or VERY MINOR ethics concerns only"]

**Limitations:**

Yes

**Paper Formatting Concerns:**

None observed.

**Quality:**

2

**Strengths And Weaknesses:**

Strengths:
1. The motivation is strong and timely. Leveraging the reasoning capabilities of MLLMs to enhance downstream generative models is a critical and underexplored direction in the V2A field. The authors convincingly demonstrate that their approach achieves state-of-the-art (SOTA) performance, validating the effectiveness of their core idea.
2. The unification of audio generation and editing within a single framework is a significant strength. By exploring two intuitive editing modalities—Region-of-Interest (ROI) based refinement and natural language instruction-based editing—the paper naturally showcases the practical benefits and versatility of incorporating an MLLM.
3. The introduction of the AudioCoT dataset is a valuable contribution in its own right. The detailed construction process and the resulting dataset will likely serve as a crucial resource for future research in reasoning-based audio generation and editing.

Weaknesses:
1. The paper lacks essential mathematical formulations. For instance, including the objective functions for fine-tuning the MLLM and for training the flow matching model would significantly improve the paper's clarity and reproducibility.
2. A more significant weakness is the lack of a dedicated performance evaluation for the MLLM's reasoning capabilities in the main text. The MLLM's ability to generate high-quality CoT is central to the entire framework's success. Therefore, its effectiveness and reliability should be explicitly demonstrated and analyzed, rather than being treated as a given. A standalone analysis of the MLLM's performance would substantially bolster the paper's core claims.

---

> ### Author Rebuttal · Authors · 2025-07-30
>
> ## Response to Reviewer 2MyN
>
> We sincerely thank the reviewer for your positive feedback and recognizing our work as `the motivation is strong and timely`, `a unified framework for audio generation and editing is a significant strength`, and `AudioCoT dataset is a valuable contribution`. Your suggestions for improving the paper's clarity and rigor are particularly valuable. Below, we hope the following point-to-point response could completely resolve your concerns. We are fully committed to open-sourcing all code, data, and models upon acceptance.
>
> ---
>
> > **1. On Mathematical Formulation**
>
> Thank you for this suggestion. We agree that providing explicit mathematical formulations is crucial for clarity and reproducibility. In the revised manuscript, we will add a **new "Preliminaries" section**. This section will formally define the problems of MLLM fine-tuning and conditional flow matching, and will present their respective loss functions and training objectives in clear mathematical terms.
>
> ---
>
> > **2. Dedicated MLLM Reasoning Evaluation**
>
> We thank the reviewer for raising this valuable suggestion. The quality of the CoT reasoning is indeed the linchpin of our framework. To further substantiate our claims and in response to your valuable suggestion, we have conducted comprehensive evaluations using both expert human annotators and automated (LLM-based) metrics:
>
> *   **Human Evaluation:**
>  We randomly sampled 100 VGGSound test set cases, covering diverse audio-visual scenarios. Two expert annotators independently rated each generated CoT across five dimensions crucial to audio-visual reasoning: (1) multimodal integration, (2) specificity of audio details, (3) feasibility for audio generation, (4) logical consistency, and (5) brevity and format compliance (≤3 sentences, ≤77 tokens). Each dimension is rated from 0 to 1 (total max 5). Disagreements >1 point were resolved by a third annotator.
>
> *   **LLM-Based Similarity Evaluation:**
>  For large-scale, objective analysis, we compared the generated CoTs to ground-truth CoTs (automatically constructed as described in Section 3.2) using GPT-4-nano, which was prompted to focus specifically on reasoning structure, causal/temporal relationships, and object-sound associations. The similarity score is on a 0–5 scale:
>
>     * 5 = Nearly identical in structure, content, and reasoning
>
>     * 4 = Mostly similar, minor omissions in details or stepwise logic
>
>     * 3 = Partially similar, significant reasoning elements missing
>
>     * 2 = Little overlap in reasoning structure/content
>
>     * 1 = Barely any overlap
>
>     * 0 = Completely unrelated
>
>
> **Results**:
>
> | Model | Human Score (0-5) | LLM CoT Similarity (0-5) |
> | :--- | :---: | :---: |
> | Qwen2.5-VL-7B | 3.78 | 3.95 |
> | Qwen2-Audio-7B | 3.82 | 4.09 |
> | **ThinkSound (VideoLLaMA2)** | **4.13** | **4.31** |
>
> These results provide **strong empirical evidence** that our fine-tuned MLLM greatly outperforms other powerful MLLMs in generating high-quality, structured CoT for V2A reasoning. This new section will be prominently featured in our revised paper.
>
> ---
>
> > **3. Inference Time Breakdown and Efficiency**
>
> We appreciate your interest in the efficiency of our framework and agree that reporting the breakdown is important for transparency and fair comparison.
>
> *   **Timing Protocol:** The reported inference time in Table 1 measures **only the online audio synthesis stage**, which is the standard practice for fair comparison in V2A generation [1, 2]. The MLLM-based CoT generation is treated as a one-time, offline pre-processing step for the entire test set, akin to feature extraction. This protocol is applied consistently across all methods. The total time for our pipeline is the sum of offline and online steps, but only the online portion is used for latency comparison. We will clarify it in the revised version.
>
> *   **Why ThinkSound is Faster:** The primary reason for ThinkSound’s superior speed is **its highly compressed VAE-based audio latents and vocoder-free architecture**, enabling much shorter sequence modeling and eliminating the heavy computational overhead of flow-based sampling and external vocoder reconstruction used in models like MMAudio.
> The detailed breakdown below clearly shows that our fast flow matching sampling and near-instant VAE decoding are the sources of our efficiency:
>
> | Method | MLLM (s) | Feature Extraction (s) | VAE/Vocoder (s) | Flow Matching Sampling (s) |
> | :--- | :---: | :---: | :---: | :---: |
> | MMAudio | – | ~1.01 | ~0.45 | ~2.56 |
> | ThinkSound | ~2.44 | ~1.20 | ~0.01 | ~1.06 |
>
> This analysis shows our pipeline separates expensive reasoning into an offline step, enabling a fast online synthesis process. We will add this breakdown to the appendix.
>
> ---
> ### [Reference]
>
> [1] Luo, Simian, et al. "Diff-foley: Synchronized video-to-audio synthesis with latent diffusion models." Advances in Neural Information Processing Systems 36 (2023): 48855-48876.
>
> [2] Wang, Yongqi, et al. "Frieren: Efficient video-to-audio generation with rectified flow matching." arXiv e-prints (2024): arXiv-2406.

---

> > ### Author Response · Authors · 2025-08-05
> >
> > We sincerely thank the reviewer for your positive feedback and constructive suggestions. Your thoughtful insights have been extremely helpful in improving our work. We have endeavored to address all your concerns comprehensively in our rebuttal and remain entirely at your disposal for any additional discussion you deem beneficial.

---

> > > ### Author Response · Authors · 2025-08-08
> > > **Gentle Reminder Regarding Remaining Discussion Time**
> > >
> > > We sincerely appreciate all the efforts and constructive feedback you have provided throughout the review process. Your valuable insights have greatly contributed to improving our work.
> > >
> > > **We believe that we have fully addressed all the questions and concerns you raised in our rebuttal. As only one day remains in the discussion period, we look forward to further discussions with you and are eager to address any new questions or concerns you might have.**
> > >
> > > Thank you once again for your time and support.

---

### Official Review · Reviewer_Ajam · 2025-07-03

**Clarity:** 3
**Significance:** 3
**Originality:** 4
**Rating:** 5
**Confidence:** 4

**Summary:**

This paper proposes ThinkSound, which is a multimodal large language model based on chain-of-thought reasoning for text,video-guided sound generation.
Through a 3-step pipeline, it can generate and edit customized sounds by borrowing knowledge from MLLM.
Moreover, it proposes AudioCoT benchmark, carefully collecting structured reasoning annotations.
It shows remarkable performance compared to the current SOTA (MMAudio) in both in-domain and out-of-distribution experiments.

**Questions:**

- The paper reports CLAP<sub>CoT</sub> scores on MovieGen Audio Bench (Table 2), suggesting that CoT reasoning captions were used. However, it is unclear whether these CoT captions were newly generated for MovieGen samples. Could the authors clarify whether reasoning-based captions were generated for MovieGen, and if so, what CLAP scores are for original captions?

- While the use of CoT reasoning appears to improve performance across metrics such as CLAP<sub>CoT</sub> and MOS, it remains unclear whether verbose or overly detailed reasoning could lead to redundant or hallucinated audio content. I suggest the authors discuss how they handle potential over-generation and whether any such cases were observed during evaluation.

- The benchmark used in Table 3 for object-focused audio generation is not clearly described. Is it based on a specific split of the AudioCoT dataset? If so, how does it differ from the test set used in Table 4? Please clarify the evaluation setup and explain any factors that may account for the differences in results.


- Are the parameter and inference time numbers in Table 1 inclusive of the VideoLLaMA2 reasoning module?


- Why does ThinkSound achieve 3× faster inference time than MMAudio, despite having slightly more parameters?

**Ethical Concerns:**

["NO or VERY MINOR ethics concerns only"]

**Final Justification:**

The authors have addressed all my concerns, so I will keep my rating for the acceptance.

**Limitations:**

yes

**Quality:**

4

**Strengths And Weaknesses:**

S1. It shows remarkable performance for TV2A.

S2. The AudioCoT benchmark would be valuable in this community.

S3. The authors provide comprehensive experiments to show the robustness and effectiveness of CoT reasoning in the TV2A task.

W1. Given the structural similarity to MMAudio, it needs to clarify the architectural advantages of the proposed method.

W2. Some parts are unclear in the experiments.

W3. More deeply failure case analysis is needed.

---

> ### Author Rebuttal · Authors · 2025-07-30
>
> ## Response to Reviewer Ajam
>
> We sincerely thank the Reviewer for the positive feedback and insightful questions. We are encouraged by your recognition of our work's strong performance and the value of our benchmark, and find your questions very helpful for clarifying our contributions. Below, we hope the point-to-point responses could completely address your concerns. We are fully committed to open-sourcing all code, models, and data upon acceptance.
>
> ---
>
> > **1. Architectural Advantages Over MMAudio**
>
> Thank you for this important question. While both MMAudio and ThinkSound leverage multimodal integration, our framework introduces several fundamental distinctions:
>
> *   **Explicit, Interpretable, and Controllable Pipeline:** ThinkSound decomposes generation into three intuitive stages: (1) foundational foley synthesis, (2) interactive object-centric refinement, and (3) instruction-guided editing. As detailed in **Sec. 4.4** and illustrated in **Fig. 1**, each stage is driven by structured CoT reasoning, making the process transparent and controllable. This is a stark contrast to MMAudio’s single-stage, black-box generation, offering users more control over the creative process. Furthermore, this design supports a richer set of conditioning, including flexible ROI inputs, audio context, global semantics, and structured CoT context (ablated in Table 5).
>
> *   **Highly Efficient Audio Latent & Faster Inference:** At the core of our efficiency lies a fully convolutional VAE achieving **32× audio compression** and enabling **direct waveform synthesis**. This design completely **obviates the need for an external vocoder**, a significant bottleneck in models like MMAudio (~0.45s overhead). This architectural choice is the primary reason ThinkSound achieves **3× faster inference** despite having a comparable parameter count, as detailed in **our response to your Question 7**.
>
> *   **Adaptive Multimodal Fusion for Enhanced Synchronization:** Our model introduces an adaptive fusion module (**Sec. 4.3**, ablated in **Table 6**) that intelligently integrates upsampled video features with audio latents via a gated mechanism. This allows the model to dynamically emphasize salient visual cues and suppress irrelevant information, leading to more precise audio-visual synchronization compared to the audio-only transformer blocks in MMAudio.
>
> We will add a dedicated paragraph in the revised manuscript to further elaborate on these crucial architectural distinctions.
>
> ---
>
> > **2. On Deeper Failure Case Analysis**
>
> We agree that a detailed failure analysis is essential. In the revised appendix, we will detail two primary failure modes with qualitative examples:
>
> *   **Limited Realism in Niche Domains:** The model struggles with highly specialized or rare sounds (e.g., specific industrial machinery and game sounds) where training data is scarce. The generated audio can sound generic or miss key acoustic characteristics, pointing to a need for more diverse training data.
>
> *   **Imprecise Spatialization for Dynamic Objects:** While our model generates stereo audio, since it is not specifically trained to constrain spatial information, it sometimes fails to accurately spatialize fast-moving or occluded objects. This is a limitation of the MLLM's ability to conduct fine-grained spatial reasoning, leading to less immersive results in complex scenes.
>
> ---
>
> > **3. CoT Caption for MovieGen and $\text{CLAP}_{\text{cap}}$ Scores**
>
> Thank you for this important question regarding our evaluation protocol. For the MovieGen Audio Bench evaluation (**Table 2**), we generated new CoT texts for each sample using our fine-tuned MLLM. The reported $\text{CLAP}_{\text{CoT}}$ scores are based on these generated texts.
>
> **For reference, $\text{CLAP}\_{\text{cap}}$ scores are (see Supplementary D.1, line 156): MMAudio 0.43, MovieGen 0.44, ThinkSound 0.49**. This shows that ThinkSound outperforms other methods with **both original and CoT-based captions**, highlighting the strength of our audio foundation model. We will clarify this in Table 2 to make it self-contained.
>
> ---
>
> > **4. Verbosity/Redundancy in CoT Reasoning**
>
> This is a crucial point. We address this risk of verbose or hallucinated CoT text through a **two-pronged strategy**: (1) rigorous, multi-stage quality control during the creation of our AudioCoT training data, and (2) direct constraints on CoT generation during inference.
>
> **First, we ensure our MLLM is trained on high-quality, concise, and factually grounded reasoning chains** through a comprehensive quality control pipeline (**Supp. A.2**), which includes automated CLAP-based filtering, LLM-based validation, and rigorous human verification, specifically:
> *   **Automated Filtering:** We use CLAP scores to automatically identify and discard audio-CoT pairs with poor semantic alignment, ensuring the MLLM learns from relevant examples.
> *   **LLM-based Validation:** We leverage GPT-4.1-nano as a validation agent to check the semantic plausibility and logical consistency of generated editing instructions, preventing illogical or "hallucinated" steps from entering our dataset.
>
> *   **Human Verification:** At each stage, a 5% random sample of the data is manually reviewed by our team. If the rejection rate exceeds a 5% threshold, we recalibrate our automated filters and reprocess the entire batch.
>
> **Second, during inference, we apply direct constraints** to the CoT generation process. Using prompts that limit reasoning length (≤ 3 sentences, ≤ 77 tokens)and focus the output strictly on audible events and their temporal relations.
>
> **Finally, to empirically validate this entire approach**, we conducted a **new ablation study** quantifying the impact of verbosity. We removed our length constraints to generate "Overly Detailed" CoT prompts and compared them against our proposed "Concise CoT".The results demonstrate a clear performance degradation with verbose reasoning:
>
> | Setting | FD↓ | **$\text{KL}_{\text{PaSST}}$**↓ | **$\text{KL}_{\text{PaNNs}}$**↓ | DeSync↓ | **$\text{CLAP}_{\text{CoT}}$**↑ |
> | :--- | :---: | :---: | :---: | :---: | :---: |
> | Overly-detailed CoT | 43.56 | 1.61 | 1.55 | 0.54 | 0.35 |
> | **Concise CoT (Ours)** | **34.56** | **1.52** | **1.32** | **0.46** | **0.46** |
>
> These results provide strong empirical evidence that **our dual strategy—proactive data curation and reactive inference constraints—is effective**. The concise reasoning fostered by our approach is critical for achieving both high objective quality and superior semantic alignment. We will add this comprehensive explanation and the new ablation study to the revised manuscript.
>
> ---
>
> > **5. Evaluation Splits and Object-Focused Benchmark**
>
> Thank you for requesting clarification. **All evaluation benchmarks are derived from the VGGSound test set but are carefully curated for different purposes**:
>
> *   **Stage 1 (Table 1 - Foundational Generation):** ~15,000 general video-audio pairs from the standard VGGSound test set.
> *   **Stage 2 (Table 3 - Object-Focused Generation):** ~2,000 samples with persistent, clearly identifiable sound-emitting objects for ROI-based evaluation.
> *   **Stage 3 (Table 4 - Audio Editing):** ~2,000 samples constructed by combining clips from distinct videos to test instruction-based editing (e.g., adding sound from one context to another). This set is distinct from the Stage 2 set.
>
> We will clarify this in the main paper, point to detailed procedures in **Supp. A.3**, and release all splits with our data.
>
> ---
>
> > **6. Model Parameter and Inference Time Reporting**
>
> Thank you for this crucial question. For fair comparison, we follow the standard protocol in prior works [1, 2, 3]. The parameter count in **Table 1 (1.30B)** refers **exclusively to the trainable parameters of our audio foundation model**, excluding all frozen components (MLLM, VAE, encoders). Correspondingly, the reported inference time **(1.07s)** measures **only the core audio synthesis pipeline** (flow matching + VAE decoding). All conditioning features (including CoT) are pre-computed offline for all methods, ensuring a direct and fair comparison of generation speed. We will make this protocol explicit in the revision.
>
> ---
>
> > **7. Why 3× Faster Inference With More Parameters?**
>
> This is an excellent observation that highlights a key strength of our architectural design. The speedup is primarily due to ThinkSound’s **efficient VAE-based audio representation (32× compression) and vocoder-free architecture**, yielding much shorter latent sequences and removing the need for a separate vocoder (which adds ~0.45s/10s audio in MMAudio). Our end-to-end pipeline thus achieves **significantly lower inference time despite comparable parameter count**. The timing breakdown below isolates this efficiency gain to the core generative steps, clearly demonstrating how our design choices lead to a substantial speedup.
>
> | Method | Flow Matching Sampling (s) | VAE/Vocoder (s) | **Total Timed (s)** |
> | :--- | :---: | :---: | :---: |
> | MMAudio | ~2.56 | ~0.45 | ~3.01 |
> | ThinkSound | ~1.06 | ~0.01 | **~1.07** |
>
> ---
> ### [Reference]
> [1] Luo, Simian, et al. "Diff-foley: Synchronized video-to-audio synthesis with latent diffusion models." Advances in Neural Information Processing Systems 36 (2023): 48855-48876.
>
> [2] Wang, Yongqi, et al. "Frieren: Efficient video-to-audio generation with rectified flow matching." arXiv e-prints (2024): arXiv-2406.
>
> [3] Cheng, Ho Kei, et al. "Taming multimodal joint training for high-quality video-to-audio synthesis." arXiv e-prints (2024): arXiv-2412.

---

> > ### Author Response · Authors · 2025-08-05
> >
> > We are deeply grateful for your accept recommendation and truly appreciate the time and thoughtful insights you have devoted to our work. Your constructive feedback has been invaluable to us. We have carefully addressed each of your concerns point-by-point in our rebuttal and would be honored to engage in any further discussion that you might find beneficial.

---

> > > ### Comment · Reviewer_Ajam · 2025-08-07
> > >
> > > Thanks to the authors for detailed answers. Most of my concerns are well discussed, so I have no more questions.

---

> > > > ### Author Response · Authors · 2025-08-07
> > > > **Follow-Up Response**
> > > >
> > > > Thank you very much for your positive feedback. We are delighted to hear that our response have addressed your concerns, and we sincerely appreciate your time and support.

---

### Author Response · Authors · 2025-08-04
**Appreciating Your Time - Welcome to Discuss**

Dear Reviewers,

Thank you for your thoughtful reviews and for taking the time to evaluate our work. We have provided our responses to your initial questions and concerns in the first phase of the rebuttal.

As we are now in the discussion phase, we wanted to let you know that we remain fully available to address any additional questions, clarify any points from our responses, or provide further details that might be helpful for your evaluation.

Please feel free to raise any follow-up questions or concerns you may have. We are committed to providing comprehensive and timely responses to ensure our work is clearly understood.

Thank you again for your valuable time and expertise.

---

### Note · Authors · 2025-08-12

Dear AC and Reviewers,

We sincerely thank the ACs for organizing the review process and the reviewers for their positive feedback and valuable suggestions. Here are the final remarks:

We were encouraged that the initial reviews showed a broad consensus on our work's value. **Our main contribution—a novel, interactive CoT-driven framework—was widely acknowledged (R-Ajam, R-2MyN, R-MPd8)**. Our experimental results were praised as **"remarkable"** (R-Ajam) and **"promising"** (R-2uba), while our carefully constructed AudioCoT dataset was unanimously recognized as a **"valuable contribution to the community"** (R-Ajam, R-2MyN, R-MPd8).

During the discussion, we actively incorporated feedback and reinforced the paper's rigor through **several key new experiments**:
1.  **Effectiveness of CoT:** In response to R-2uba and R-Ajam, a new ablation study confirmed that our T5 encoder effectively leverages the "ordered logical structure" of CoT, not just keywords. This directly addresses concerns about the method's robustness.
2.  **MLLM Reasoning Evaluation:** As suggested by R-2MyN, we conducted a comprehensive human and automated evaluation of our CoT generation module, demonstrating that our fine-tuned MLLM substantially outperforms baselines in generating high-quality V2A reasoning chains.
3.  **Architectural Validation:** To address R-MPd8's concern, we added a new ablation on the full MM-DiT architecture, demonstrating the critical role of its multi-stream fusion design.

We are grateful that these discussions were highly productive. All engaging reviewers explicitly confirmed that their concerns were well-resolved. R-Ajam noted, `Most of my concerns are well discussed,` and R-MPd8 stated, `My concerns are well addressed.` R-2uba also confirmed, `The problem tacked is solid. The effort in paper and rebuttal solid.` **Crucially, based on these clarifications and new experiments, both Reviewer 2uba and Reviewer MPd8 raised their scores to a positive value, strongly demonstrating the soundness of our work and the effectiveness of our rebuttal.**

We are confident that our paper is now more complete and rigorous. ThinkSound offers a novel, interpretable, and reasoning-driven paradigm for video-to-audio generation, and we hope it will be a valuable contribution to the community. We **commit to open-sourcing all code, models, and the AudioCoT dataset** to foster future research and reproducibility.

Thank you again for your time and consideration.

---

### Decision · Program_Chairs · 2025-09-17

**Decision:**

Accept (poster)

**Comment:**

**Scientific claims:**
The paper introduces a reasoning-driven framework for video-to-audio (V2A) generation based on Chain-of-Thought (CoT) reasoning from an MLLM. It proposes object-centric refinement and instruction-based editing. The authors also contribute a dataset  (AudioCoT dataset) of structured reasoning annotations. Experiments on VGGSound and MovieGen show strong results.


**Weaknesses:**   Concerns about robustness and qualitative evaluation depth.

**Decision:** Accept (poster). Authors adequately addressed most concerns; reviewers agreed, as evidenced by raised scores confirming resolution.

**During the discussion**, The Authors added key ablations to clarify concerns: the T5 encoder uses CoT order, not just keywords (Ajam, 2uba); human + automated MLLM reasoning evaluations show clear gains (2MyN); and MM-DiT fusion ablation confirms its role (MPd8).
Reviewers acknowledged that concerns were resolved.